# Bowel movement frequency and risks of major vascular and non-vascular diseases: a population-based cohort study among Chinese adults

Songchun Yang,[1] Canqing Yu,[1] Yu Guo,[2] Zheng Bian,[2] Mengyu Fan,[1] Ling Yang,[3,4] Huaidong Du,[3,4] Yiping Chen,[3,4] Shichun Yan,[5] Yajing Zang,[6] Junshi Chen,[7] Zhengming Chen,[4] Jun Lv  ,[1,8,9] Liming Li,[1] on behalf of the China Kadoorie Biobank Collaborative Group

For numbered affiliations see end of article.

**Correspondence to**
Professor Jun Lv;
lvjun@bjmu.edu.cn

## ABSTRACT

**Objective** The application of bowel movement frequency (BMF) in primary care is limited by the lack of solid evidence about the associations of BMF with health outcomes apart from Parkinson's disease and colorectal cancer. We examined the prospective associations of BMF with major vascular and non-vascular diseases outside the digestive system.

**Design** Population-based prospective cohort study.

**Setting** The China Kadoorie Biobank in which participants from 10 geographically diverse areas across China were enrolled between 2004 and 2008.

**Participants** 487 198 participants aged 30 to 79 years without cancer, heart disease or stroke at baseline were included and followed up for a median of 10 years. The usual BMF was self-reported once at baseline.

**Primary and secondary outcome measures** Incident events of predefined major vascular and non-vascular diseases.

**Results** In multivariable-adjusted analyses, participants having bowel movements 'more than once a day' had higher risks of ischaemic heart disease (IHD), heart failure, chronic obstructive pulmonary disease, type 2 diabetes mellitus and chronic kidney disease (CKD) when compared with the reference group ('once a day'). The respective HRs (95% CIs) were 1.12 (1.09 to 1.16), 1.33 (1.22 to 1.46), 1.28 (1.22 to 1.36), 1.20 (1.15 to 1.26) and 1.15 (1.07 to 1.24). The lowest BMF ('less than three times a week') was also associated with higher risks of IHD, major coronary events, ischaemic stroke and CKD. The respective HRs were 1.07 (1.02 to 1.12), 1.22 (1.10 to 1.36), 1.11 (1.05 to 1.16) and 1.20 (1.07 to 1.35).

**Conclusion** BMF was associated with future risks of multiple vascular and non-vascular diseases. The integration of BMF assessment and health counselling into primary care should be considered.

## Strengths and limitations of this study

- ► In this large prospective cohort study of Chinese adults, bowel movement frequency (BMF) was associated with risks of multiple vascular and non-vascular diseases outside the digestive system.
- ► The results highlight that gastrointestinal health is related to the health of multiple systems and the integration of BMF assessment and health counselling into primary care practice should be considered.
- ► The prospective cohort design, large sample size and long-term follow-up allowed the present study to produce reliable risk estimates.
- ► BMF was self-reported once at baseline and lack of other detailed information about intestinal health.
- ► Residual confounding may still exist, such as total energy intake, laxative use and other non-cardiovascular medications that might directly affect BMF.

(CVDs),[1–3] type 2 diabetes mellitus (T2DM)[4] and chronic kidney disease (CKD).[5] Recent advancements in gut microbiota have also improved our understanding of the potential relationship of their composition, diversity and metabolites with diseases.[6–10] Bowel movement is the endpoint of the digestion process. Bowel movement frequency (BMF) may serve as a simple indicator of adequate colonic function[11] and may reflect lifestyle factors (eg, diet and exercise)[12 13] and individual characteristics (eg, gut microbiota).[14 15] It could inform the prevention and early detection of diseases.

A meta-analysis including four cohort and five case-control studies has reported that constipation is associated with subsequent diagnosis of Parkinson's disease.[16] Our recent study has shown that higher BMF is associated with an increased risk of colorectal cancer

## INTRODUCTION

A properly functioning gastrointestinal tract plays an essential role in health. Diseases of the digestion system have been prospectively associated with cardiovascular diseases

(CRC) only in the first 5 years of follow-up, indicating that abnormal increase of BMF may be an early symptom of CRC.[17] To our knowledge, only five cohort studies conducted in the USA and Japan have examined the associations between BMF and CVDs, but findings remained inconclusive.[18–22] For other outcomes, only one retrospective cohort study conducted in the USA has shown that constipation was associated with higher risks of incident CKD and end-stage renal disease.[23] More evidence from large population-based cohort studies is warranted before such a message that abnormal BMF may indicate future disease risks could be applied to primary care practice.

Therefore, we prospectively examined the associations of BMF with major vascular and non-vascular diseases outside the digestive system in the China Kadoorie Biobank (CKB) study of 0.5 million adults.

## SUBJECTS AND METHODS
### Study population
CKB is an ongoing large prospective cohort study of over 0.5 million adults from 10 geographically diverse sites across China. The baseline survey was conducted during 2004 to 2008 and the follow-up was initiated soon afterward. Details of the study have been described previously.[24 25] Briefly, a total of 512 715 adults aged 30 to 79 years had valid baseline data including a complete interviewer-administered laptop-based questionnaire, physical measurements and a written informed consent form.

In the main analyses, we excluded participants: (i) who had a previous diagnosis of cancer (n=2578), heart disease (n=15 472) or stroke (n=8884) at baseline; (ii) who were lost to follow-up soon after baseline (n=1) and (iii) who had missing body mass index (BMI; n=2). After these exclusions, 487 198 eligible participants remained. In the analyses of specific diseases, we additionally excluded participants: (i) who had a prior diagnosis of emphysema, bronchitis or pulmonary heart disease (n=13 288), and who had airflow obstruction (AFO)[26] diagnosed by spirometry at baseline (n=23 767) (for chronic obstructive pulmonary disease (COPD)); (ii) who had a self-reported (n=16 162) or screen-detected (n=14 138) diabetes at baseline (for T2DM) and (iii) who had a prior diagnosis of kidney disease (n=7575) (for CKD).

### Assessment of exposure
BMF was assessed at the baseline survey by the question 'About how often do you have bowel movements each week?' The four response options were more than once a day, once a day, once every 2 to 3 days or less than three times a week. After completion of the baseline survey in July 2008, we randomly selected about 5% of the surviving participants for a resurvey in the same year. To test the reproducibility of the BMF question, we included 1300 participants who completed the same questionnaire twice at an interval of fewer than 1.5 years (median 1.4 years). The linearly and quadratically weighted kappa coefficients[27] between the two questions were 0.40 and 0.46, respectively.

### Assessment of covariates
The baseline questionnaire collected covariates information on sociodemographic characteristics, lifestyle behaviours, dietary habits, personal health and family medical history. The daily level of physical activity was calculated by multiplying the metabolic equivalent of tasks (METs) value for a particular type of physical activity by hours spent on that activity per day and summing the MET-hours for all activities.[28] A short validated qualitative food frequency questionnaire with five categories of frequency (daily, 4 to 6 days per week, 1 to 3 days per week, monthly or never or rarely) was used to collect information on 12 major food groups during the previous 12 months (see details in the online supplementary file).[29] The amounts of consumption were not collected.

Height, body weight and waist circumference (WC) were measured with calibrated instruments. BMI was calculated as the measured weight in kilograms divided by the square of the measured height in metres. Prevalent diabetes at baseline was defined as self-reported diabetes diagnosed by a physician before the baseline survey or screen-detected diabetes, which was defined as[30] (i) random blood glucose ≥7.0 mmol/L and fasting time ≥8 hour, (ii) random blood glucose ≥11.1 mmol/L and fasting time <8 hour or (iii) fasting blood glucose ≥7.0 mmol/L on subsequent testing. Forced expiratory volume in 1 s (FEV1) and forced vital capacity (FVC) were measured using a hand-held Micro (MS01) Spirometer (CareFusion Corp). Screen-detected AFO was defined as pre-bronchodilator FEV1/FVC <0.70.[26]

### Assessment of outcomes
All participants were followed-up for incident disease outcomes since their enrolment into the study at baseline. Incident diseases were identified by using linkages with local disease and death registries and national health insurance (HI) system, and by active follow-up.[25] Until the end of 2016, nearly 97% participants had been linked to the HI databases, with similar proportions of successful linkage for participants across the 10 survey sites. Trained staff blinded to baseline information coded all events using International Classification of Diseases, Tenth Revision.

The study outcomes in the present analyses included a range of prespecified vascular and non-vascular diseases: (i) ischaemic heart disease (IHD) (I20-I25), (ii) major coronary events (MCEs) (including non-fatal I21-I23 and fatal I21-I25), (iii) heart failure (I50), (iv) haemorrhagic stroke (I61), (v) ischaemic stroke (I63), (vi) COPD (J41-J44), (vii) T2DM (E11, E14), (viii) CKD: including diabetes mellitus with renal complications (E10.2, E11.2, E12.2, E13.2 and E14.2), hypertensive renal disease (I12 and I13), glomerular disease (N03, N04, N05 and N07), renal tubulointerstitial disease (N11, N12, N13, N14 and N15) and renal failure (N18 and N19).

## Statistical analysis

The mean and prevalence values of baseline characteristics were calculated across different BMF categories by using analyses of covariance for continuous variables and logistic regressions for categorical variables, adjusting for age (years), sex and 10 survey sites where appropriate. Person-years at risk were calculated from the baseline date to the diagnosis of the study outcome of interest, death, loss to follow-up or 31 December 2016, whichever came first.

Stratified Cox proportional hazards models were used to estimate adjusted HRs and 95% CIs, with age as the underlying time scale and stratified jointly by 10 survey sites and age at baseline (5-year intervals). The proportional hazards assumption for the Cox models was checked by log-log plots; no apparent violation was found. The covariates adjusted in the multivariable models were determined according to previous literatures and biological relevance.[12 13 18–23] The covariates included sex (for the whole cohort analysis); education; occupation; household income; marital status; family history of certain diseases (for corresponding disease analysis only); smoking status; total physical activity level; alcohol consumption; intake frequency of fresh vegetables, fresh fruits and red meat; BMI; waist circumference and prevalent hypertension and diabetes at baseline (no adjustment for T2DM). The linear trend test of disease risks across various metrics of BMF was performed by modelling the levels of ordered categorical variables as a continuous variable in a separate model.

In sensitivity analyses, we excluded cases that occurred during the first 2 years of follow-up. We also further adjusted for the following factors one by one as potential confounders: intake frequency of rice, wheat, other staple food, spicy food and tea; consumption of nutritional supplements (fish oil/cod liver oil, vitamins, calcium/iron/zinc, ginseng and other herbal products); medications for CVDs (aspirin, angiotensin-convertingenzyme inhibitors, beta-blocker, statins, diuretics and calcium antagonist); medical history of kidney disease, peptic ulcer, cirrhosis or chronic hepatitis and gallstone or gallbladder disease at baseline.

To explore whether the associations of BMF with the study outcomes differed by baseline characteristics, we stratified the analyses by the following factors: sex (men or women), age (<60 year, ≥60 year), hypertension (presence or absence), obesity status defined jointly by BMI and WC (presence: BMI≥28.0 kg/m² or WC ≥90.0 cm (men)/≥85.0 cm (women);[31] absence). The definition of obesity status was used because of the significance of abdominal obesity for the rapid growth of disease risk in Chinese adults.[32] The tests for multiplicative interaction were performed using likelihood ratio tests by comparing models with and without cross-product terms between the stratifying variable and BMF categories.

P values are presented as unadjusted for multiple testing unless otherwise indicated. For testing of multiple primary outcomes, a Bonferroni correction was applied to the significance level, α=0.05 (two-sided), that divided 0.05 by the number of outcomes examined (five vascular and three non-vascular outcomes). The analyses were conducted using Stata (V.15.0, StataCorp).

## Patient and public involvement

Patients were not involved in the present study. The results of the main study were presented to study participants at the website of the CKB study (http://www.ckbiobank.org/site/) and by newsletters annually.

## RESULTS

### Baseline characteristics of study participants

The distributions of sociodemographic factors, lifestyle factors and health status at baseline across four categories of BMF are displayed in table 1. Of the 487 198 participants analysed, the mean age was 51.5 years, 59.1% were women and 43.1% were from urban areas. Participants who defecated more frequently were more likely to be men and rural residents. They were also more likely to have higher BMI and WC, and to have hypertension. Men who drank alcohol regularly were more likely to have frequent bowel movements.

### Associations of BMF with risks of multiple vascular and non-vascular diseases

During a median follow-up of 10.1 years (IQR 2.0 years), we documented 40 347 IHD, 7972 MCEs, 8943 haemorrhagic strokes, 37 579 ischaemic strokes, 4204 heart failure, 11 054 COPD, 15 281 T2DM and 6526 CKD. In the whole cohort, after multivariable adjustment, participants who reported having bowel movements 'more than once a day' were associated with higher risks of IHD, heart failure, COPD, T2DM and CKD when compared with the reference group ('once a day'). The corresponding HRs (95% CIs) were 1.12 (1.09 to 1.16), 1.33 (1.22 to 1.46), 1.28 (1.22 to 1.36), 1.20 (1.15 to 1.26) and 1.15 (1.07 to 1.24), respectively (all p values <0.00625; table 2). At the same time, the lowest BMF ('less than three times a week') was associated with increased risks of IHD, MCEs, ischaemic stroke and CKD. The corresponding HRs (95% CIs) were 1.07 (1.02 to 1.12), 1.22 (1.10 to 1.36), 1.11 (1.05 to 1.16) and 1.20 (1.07 to 1.35), respectively (all p values <0.00625). We observed linear trends between BMF and the risks of certain outcomes, with positive associations for heart failure, COPD and T2DM, and negative associations for MCEs and ischaemic stroke (all p values for trend <0.00625) (table 2). Results of stepwise adjusted models for each disease outcome are presented in online supplementary table 1. The association between BMF and T2DM attenuated after additional adjustment for BMI and waist circumference.

### Sensitive analyses

In the sensitivity analyses, the associations of BMF with multiple disease outcomes did not change appreciably after excluding cases that occurred in the first

**Table 1** Baseline characteristics according to BMF at baseline for 487 198 participants

| | More than once a day | Once a day | Once every 2 to 3 days | Less than three times a week |
|---|---|---|---|---|
| Number of participants, n (%) | 46 426 (9.5) | 373 054 (76.6) | 46 570 (9.6) | 21 148 (4.3) |
| Women, % | 45.9 | 57.7 | 72.6 | 82.5 |
| Age, year | 53.0 | 51.5 | 50.6 | 51.0 |
| Urban area, % | 33.3 | 43.9 | 43.7 | 49.4 |
| Middle school and above, % | 46.7 | 49.5 | 49.1 | 48.4 |
| Agricultural and industrial workers, % | 58.2 | 57.4 | 57.9 | 57.5 |
| Household income ≥20 000 RMB/year, % | 41.8 | 43.4 | 40.6 | 38.8 |
| Married, % | 91.4 | 91.0 | 90.2 | 89.2 |
| Current tobacco smoker, % | | | | |
| Men | 68.2 | 67.5 | 70.4 | 72.5 |
| Women | 2.6 | 2.7 | 2.8 | 3.2 |
| Current daily alcohol drinker, % | | | | |
| Men | 25.4 | 21.0 | 15.3 | 14.8 |
| Women | 1.0 | 1.0 | 0.8 | 0.7 |
| Physical activity, MET-hours/day | 22.0 | 21.6 | 21.3 | 20.9 |
| Regular consumption of*, % | | | | |
| Red meat | 47.0 | 47.4 | 46.9 | 45.3 |
| Fresh vegetables | 98.5 | 98.2 | 98.5 | 98.2 |
| Fresh fruits | 27.2 | 28.1 | 26.7 | 23.5 |
| Body mass index, kg/m² | 24.5 | 23.6 | 22.8 | 22.6 |
| Waist circumference, cm | 82.4 | 80.2 | 77.9 | 77.3 |
| Hypertension, % | 38.8 | 34.2 | 27.5 | 25.7 |
| Diabetes, % | 6.0 | 5.2 | 5.5 | 6.4 |
| Kidney disease†, % | 1.9 | 1.4 | 1.3 | 1.4 |
| COPD†, % | 8.5 | 6.9 | 7.0 | 7.7 |
| Family history of, % | | | | |
| Cancer | 18.0 | 16.4 | 16.6 | 17.2 |
| Diabetes | 8.3 | 6.7 | 6.7 | 6.8 |
| Heart attack | 3.6 | 3.1 | 3.1 | 3.6 |
| Stroke | 19.4 | 17.5 | 17.0 | 17.2 |

Baseline characteristics were presented as mean or percentage. Age and sex were not adjusted, and others were adjusted for age, sex and survey sites as appropriate. Linear trend was assessed by assigning consecutive integers to four BMF categories in a separate model. All p values for trend were <0.001, except for agricultural and industrial workers (p=0.359), regular consumption of fresh vegetables (p=0.955), the prevalence of diabetes (p=0.024), family history of cancer (p=0.010) and heart attack (p=0.486).
*Regular consumption was defined as consumption at least 4 to 6 days per week.
†Participants with kidney disease or COPD were further excluded from the analyses of the corresponding outcome.
BMF, bowel movement frequency; COPD, chronic obstructive pulmonary disease; MET, metabolic equivalent of task.

2 years of follow-up. The HRs (95% CIs) were almost unchanged after additional adjustment for other potential confounders (data not shown).

## Subgroup analyses

There was a significant interaction between BMF and obesity status on the risk of MCEs (p for interaction=0.001). Compared with participants with BMF of once every 1 to 3 days, the increased risk of MCEs associated with lower BMF was only observed among obese participants (table 3). There was no heterogeneity by sex in the association between BMF and most disease outcomes except for ischaemic stroke (p for interaction=0.006) (online supplementary table 2). The associations between BMF and all study outcomes were consistent across different age groups or hypertension status. (online supplementary table 3 and online supplementary table 4).

## DISCUSSION

In this large prospective cohort study of Chinese adults, BMF was associated with risks of multiple vascular and non-vascular diseases outside the digestive system independent of traditional lifestyle and intermediate risk factors for common non-communicable diseases. Compared with BMF of once a day, low BMF (<3 times/week) was associated with increased risks of MCEs and ischaemic stroke; more frequent bowel movements were associated with increased risks of heart failure, COPD and T2DM. Increased risks of IHD and CKD were observed in participants who were either constipated or had frequent bowel movements. BMF was not associated with the risk of haemorrhagic stroke. Our findings persisted after excluding cases that occurred in the first 2 years of follow-up.

Several prospective studies have examined the associations of BMF with CVDs.[18–21] The Women's Health Initiative observational study followed 73 047 postmenopausal women for 6.4 years and assessed the severity of constipation by self-report at baseline.[18] In that study, women with severe constipation had a 23% higher risk of death from a composite of cardiovascular events (HR=1.23; 95% CI: 1.03 to 1.47) compared with women with no constipation. Separate analysis for each type of cardiovascular events was limited by the small number of events. Another retrospective cohort study conducted in over three million US veterans (93.2% of men) showed similar results.[22] In that study, constipation was defined as either having ≥2 prescriptions for ≥30 day supply of laxatives each or having ≥2 diagnoses for constipation identified by International Classification of Diseases, Ninth Revision, Clinical Modification. Compared with participants without constipation, those with constipation had higher risks of incident coronary heart disease (HR=1.11; 95% CI: 1.08 to 1.14) and ischaemic stroke (HR=1.19; 95% CI:1.15 to 1.22). However, findings from the Nurse's Health Study of 86 289 women (aged 30 to 55 years, followed-up ~30 years) showed no associations of BMF with incident coronary heart disease, incident stroke and cardiovascular mortality.[21] Compared with women with daily bowel movements, those with BMF >1 time/day had a modest increase in the risk of total mortality (HR=1.10; 95% CI: 1.06 to 1.15).

**Table 2**  HRs (95% CIs) for associations between BMF and multiple vascular and non-vascular diseases

| Diseases | More than once a day | Once a day | Once every 2 to 3 days | Less than three times a week | P value for trend* |
|---|---|---|---|---|---|
| **Vascular diseases** | | | | | |
| **IHD** | | | | | |
| No. of cases | 4668 | 30 470 | 3448 | 1761 | |
| Cases/PYs (1/1000) | 10.43 | 8.45 | 7.60 | 8.59 | |
| Multivariable adjusted | 1.12 (1.09 to 1.16) | 1.00 | 1.00 (0.96 to 1.03) | 1.07 (1.02 to 1.12) | 0.008 |
| **MCEs** | | | | | |
| No. of cases | 871 | 5960 | 765 | 376 | |
| Cases/PYs (1/1000) | 1.88 | 1.61 | 1.64 | 1.78 | |
| Multivariable adjusted | 1.03 (0.96 to 1.11) | 1.00 | 1.10 (1.02 to 1.19) | 1.22 (1.10 to 1.36) | 0.002 |
| **Haemorrhagic stroke** | | | | | |
| No. of cases | 1028 | 6856 | 723 | 336 | |
| Cases/PYs (1/1000) | 2.22 | 1.85 | 1.56 | 1.60 | |
| Multivariable adjusted | 0.97 (0.90 to 1.03) | 1.00 | 0.98 (0.91 to 1.06) | 1.07 (0.95 to 1.19) | 0.256 |
| **Ischaemic stroke** | | | | | |
| No. of cases | 4044 | 28 594 | 3291 | 1650 | |
| Cases/PYs (1/1000) | 9.00 | 7.91 | 7.24 | 8.03 | |
| Multivariable adjusted | 1.01 (0.98 to 1.05) | 1.00 | 1.04 (1.00 to 1.08) | 1.10 (1.05 to 1.16) | 0.002 |
| **Heart failure** | | | | | |
| No. of cases | 535 | 3068 | 398 | 203 | |
| Cases/PYs (1/1000) | 1.16 | 0.83 | 0.86 | 0.96 | |
| Multivariable adjusted | 1.33 (1.22 to 1.46) | 1.00 | 0.96 (0.86 to 1.06) | 0.99 (0.86 to 1.15) | <0.001 |
| **Non-vascular diseases** | | | | | |
| **COPD** | | | | | |
| No. of cases | 1518 | 8167 | 926 | 443 | |
| Cases/PYs (1/1000) | 3.63 | 2.37 | 2.14 | 2.29 | |
| Multivariable adjusted | 1.28 (1.22 to 1.36) | 1.00 | 0.89 (0.83 to 0.95) | 0.92 (0.84 to 1.02) | <0.001 |
| **T2DM** | | | | | |
| No. of cases | 2207 | 11 514 | 1077 | 483 | |
| Cases/PYs (1/1000) | 5.14 | 3.30 | 2.46 | 2.46 | |
| Multivariable adjusted | 1.20 (1.15 to 1.26) | 1.00 | 0.86 (0.81 to 0.92) | 0.84 (0.77 to 0.92) | <0.001 |
| **CKD** | | | | | |
| No. of cases | 821 | 4786 | 605 | 314 | |
| Cases/PYs (1/1000) | 1.81 | 1.31 | 1.33 | 1.52 | |
| Multivariable adjusted | 1.15 (1.07 to 1.24) | 1.00 | 1.07 (0.98 to 1.17) | 1.20 (1.07 to 1.35) | 0.577 |

Cox regression models were stratified by 5 year groups and 10 survey sites, with age as the underlying time scale.

Multivariable models were adjusted for sex (men or women); level of education (no formal school, primary school, middle school, high school, college or university or higher); occupation (agricultural, industrial, administrative or managerial, professional or technical, sales or service, retired, house wife or husband, self-employed, unemployed, other); household income (<2500, 2500 to 4999, 5000 to 9999, 10 000 to 19 999, 20 000 to 34 999, ≥35 000 RMB per year); marital status (married, widowed, divorced or separated or never married); family history of certain diseases (presence, absence or unclear; for corresponding disease only; no adjustment for heart failure, COPD and CKD); smoking status (non-smoker, former smoker, current smoker who smoked <15, 15 to 24 or ≥25 cigarettes or equivalents per day; those who had stopped smoking because of ill health were included in current smokers to avoid bias); total physical activity level (MET-hours/day); alcohol consumption (non-drinker, former weekly drinker, weekly drinker, daily drinker with an intake of <15, 15 to 29, 30 to 59 or ≥60 g/day of pure alcohol); intake frequency of fresh vegetables, fresh fruit and red meat (daily, 4 to 6 days/week, 1 to 3 days/week, monthly or rarely or never); BMI (kg/m2); waist circumference (cm); prevalent hypertension and diabetes at baseline (presence or absence, no adjustment for T2DM).

*Assessed by assigning consecutive integers to four BMF categories in a separate model. A Bonferroni corrected threshold was used, α=0.00625.

BMF, bowel movement frequency;BMI, body mass index; CKD, chronic kidney disease; COPD, chronic obstructive pulmonary disease; IHD, ischaemic heart disease; MCEs, major coronary events;PYs, person-years; T2DM, type 2 diabetes mellitus.

**Table 3** HRs (95% CIs) for associations between BMF and multiple vascular and non-vascular diseases stratified by obesity status

| Diseases | Obesity subgroups | More than once a day | | Once every 1 to 3 days* | | Less than three times a week | | P value for interaction† |
|---|---|---|---|---|---|---|---|---|
| | | Cases | HR (95%CI) | Cases | HR (95%CI) | Cases | HR (95%CI) | |
| Vascular diseases | | | | | | | | |
| IHD | | | | | | | | 0.383 |
| | No | 2771 | 1.15 (1.10 to 1.19) | 22 109 | 1 | 1258 | 1.04 (0.98 to 1.11) | |
| | Yes | 1897 | 1.10 (1.04 to 1.15) | 11 809 | 1 | 503 | 1.13 (1.03 to 1.23) | |
| MCEs | | | | | | | | 0.001 |
| | No | 577 | 1.07 (0.98 to 1.17) | 4478 | 1 | 238 | 1.07 (0.93 to 1.22) | |
| | Yes | 294 | 0.91 (0.81 to 1.03) | 2247 | 1 | 138 | 1.57 (1.31 to 1.86) | |
| Haemorrhagic stroke | | | | | | | | 0.624 |
| | No | 706 | 0.98 (0.90 to 1.06) | 5543 | 1 | 269 | 1.08 (0.96 to 1.23) | |
| | Yes | 322 | 0.93 (0.82 to 1.04) | 2036 | 1 | 67 | 1.05 (0.82 to 1.34) | |
| Ischaemic stroke | | | | | | | | 0.688 |
| | No | 2367 | 1.02 (0.98 to 1.07) | 20 903 | 1 | 1221 | 1.10 (1.03 to 1.16) | |
| | Yes | 1677 | 1.01 (0.96 to 1.06) | 10 982 | 1 | 429 | 1.07 (0.97 to 1.18) | |
| Heart failure | | | | | | | | 0.356 |
| | No | 360 | 1.36 (1.22 to 1.52) | 2493 | 1 | 152 | 0.96 (0.81 to 1.13) | |
| | Yes | 175 | 1.25 (1.06 to 1.47) | 973 | 1 | 51 | 1.22 (0.92 to 1.62) | |
| Non-vascular diseases | | | | | | | | |
| COPD | | | | | | | | 0.353 |
| | No | 1132 | 1.29 (1.21 to 1.37) | 7070 | 1 | 371 | 0.98 (0.88 to 1.09) | |
| | Yes | 386 | 1.21 (1.08 to 1.35) | 2023 | 1 | 72 | 0.90 (0.71 to 1.14) | |
| T2DM | | | | | | | | 0.506 |
| | No | 1048 | 1.35 (1.27 to 1.44) | 7069 | 1 | 338 | 0.80 (0.72 to 0.90) | |
| | Yes | 1159 | 1.25 (1.17 to 1.33) | 5522 | 1 | 142 | 0.77 (0.65 to 0.91) | |
| CKD | | | | | | | | 0.051 |
| | No | 545 | 1.22 (1.11 to 1.33) | 3789 | 1 | 252 | 1.23 (1.08 to 1.40) | |
| | Yes | 276 | 1.01 (0.88 to 1.15) | 1602 | 1 | 62 | 1.05 (0.81 to 1.35) | |

Multivariable models adjusted for the same set of covariates as table 2, except for BMI (kg/m$^2$) and waist circumference (cm). Obesity status was defined according to BMI and WC, with 'yes' as BMI ≥28.0 kg/m$^2$ or WC ≥90.0 cm (men) / ≥85.0 cm (women).
*Combined group including 'once a day' and 'once every 2 to 3 days'.
†A Bonferroni corrected threshold was used, α=0.00625.
BMF, bowel movement frequency; BMI, body mass index; CKD, chronic kidney disease; COPD, chronic obstructive pulmonary disease; IHD, ischaemic heart disease; MCEs, major coronary events; T2DM, type 2 diabetes mellitus; WC, waist circumference.

The Ohsaki cohort including 45 112 Japanese aged 40 to 79 years (follow-up 13.3 years) showed that the risk of CVD mortality increased in participants reporting BMF of 1 time/2 to 3 days (HR=1.21; 95%: 1.08 to 1.35) and ≤1 time/4 days (HR=1.39; 95% CI: 1.06 to 1.81) compared with those reporting ≥1 time/day.[19] The respective HRs (95% CI) for death from ischaemic stroke were 1.27 (1.00 to 1.61) and 1.97 (1.21 to 3.21). The study was underpowered for analyses of IHD and stroke mortality of other subtypes. Another similar study based on the Japan Collaborative Cohort of 72 014 Japanese aged 40 to 79 years did not observe significant associations of BMF with deaths from coronary heart disease, ischaemic and haemorrhagic stroke, but it was also underpowered to allow a firm conclusion.[20] However, this study indicated that laxative use, a potential reflection of severe constipation, was associated with increased risks of deaths from coronary heart disease and ischaemic stroke.

In the present study, we showed that low BMF was associated with increased risks of MCEs and ischaemic stroke, in line with findings from most previous studies. With regard to the associations between BMF and other outcomes, only one retrospective cohort study conducted in US veterans examined the association between constipation and incident CKD.[23] In that study, veterans with constipation had higher risks of CKD (HR=1.13; 95% CI: 1.11 to

1.14) and end-stage renal disease (HR=1.09; 95% CI: 1.01 to 1.18) compared with those without constipation.

Previous studies have linked constipation or high frequency of bowel movements to the alterations of composition, diversity and functions of gut microbiota.[15 33 34] Altered gut microbiota has been suggested to interact with the host through both metabolism dependent and independent processes and contribute to the development and progression of diseases such as atherosclerosis, heart failure, CKD and T2DM.[35] Similarly, the gastrointestinal tract and respiratory tract, although separate organs, are part of a shared mucosal immune system (ie, the gut-lung axis).[36] The microbiota and metabolites in the gut and the lungs can modulate systemic and local immunity and influence the pathogenesis of respiratory diseases.[36] Also, slow transit constipation is accompanied by reduced fermentation efficacy of complex carbohydrates, which diminishes the production of short-chain fatty acids, an energy source for colonocytes, leading to oxidative stress.[37] Oxidative stress has also been associated with the onset and progression of atherosclerosis, CVDs, neurological diseases and kidney diseases.[38] Although several plausible mechanisms underlying these associations have been suggested, further research is warranted to provide convincing evidence of the causal connection.

Dietary factors might influence BMF. Previous studies in CKB have shown that consumption of fresh fruits, fresh eggs and spicy food was each associated with lower risks of CVDs[39–41] and diabetes.[42] In the present study, the relative risk estimates for multiple disease outcomes did not change appreciably after adjusting for these dietary factors, indicating that the associations between BMF and multiple diseases might be independent of dietary factors. Other established risk factors for chronic diseases that may affect BMF, such as physical inactivity and obesity,[12 13] did not confound these associations either, except for obesity in the analysis of T2DM. Besides, most of the associations between BMF and multiple disease outcomes were consistent across baseline subgroups defined by sex, age, obesity status and hypertension status in the present analyses. The significant increase in the risk of MCEs associated with constipation was only observed among obese participants. We assume that both obesity-related and constipation-related oxidative stress act synergistically to increase the risk of MCEs. There is still a limited understanding of the sex difference in the association between constipation and ischaemic stroke.

To our knowledge, this is the first prospective study examining the associations of BMF with a range of disease outcomes outside the digestive system collected through linkages to multiple data systems. The present study had a large sample size and long-term follow-up, and hence was well powered for analyses of CVD subtypes and other outcomes. Participants included both men and women geographically spread across urban and rural areas across China. Prospective design and long-term follow-up minimised the potential for reverse causation. A detailed collection of lifestyle factors and other covariates allowed adjustment for several potential confounders.

However, there are several limitations. First, BMF was self-reported, which may cause non-differential misclassification and potentially attenuate the true associations. BMF and covariate information collected once at baseline might have changed during the follow-up. Second, although we controlled for several potential confounders, residual confounding may still exist. For example, we were not able to adjust for total energy intake, laxative use, opioids use and other non-cardiovascular medications that might directly affect BMF. Third, due to the lack of detailed information about intestinal health (eg, colonic transit time, stool consistency, gut microbiota and blood biomarkers), there is a limited understanding of the relationship between the gut and multiple diseases.

In conclusion, we observed that BMF was associated with future risk of multiple vascular and non-vascular diseases, including CVD, COPD, T2DM and CKD in this Chinese population. Our findings highlight that gastrointestinal health is related to the health of multiple systems and should stimulate future interest in elucidating the mechanisms underlying these associations. BMF is an easily assessed indicator and has been correlated with colonic transit time, an indicator of gastrointestinal function that is more accurate but not easy for use.[11] Based on the findings of the current and previous studies,[16 17] people with abnormal BMF should consider the possibility of undiagnosed diseases and be aware of the potential future risks of various health conditions. The integration of BMF assessment and health counselling into primary care practice should be considered. Further studies are needed to evaluate factors that may influence BMF to improve and maintain digestive health.

**Author affiliations**
[1]Department of Epidemiology and Biostatistics, School of Public Health, Peking University Health Science Centre, Beijing, China
[2]Chinese Academy of Medical Sciences, Beijing, China
[3]Medical Research Council Population Health Research Unit, Nuffield Department of Population Health, University of Oxford, Oxford, United Kingdom
[4]Clinical Trial Service Unit and Epidemiological Studies Unit, Nuffield Department of Population Health, University of Oxford, Oxford, United Kingdom
[5]NCDs Prevention and Control Department, Heilongjiang Centre for Disease Control and Prevention, Harbin, Heilongjiang, China
[6]Qingdao Centre for Disease Control and Prevention, Qingdao, Shandong, China
[7]China National Centre for Food Safety Risk Assessment, Beijing, China
[8]Key Laboratory of Molecular Cardiovascular Sciences (Peking University), Ministry of Education, Beijing, China
[9]Peking University Institute of Environmental Medicine, Peking University, Beijing, China

**Acknowledgements** The most important acknowledgment is to the participants in the study and the members of the survey teams in each of the 10 regional centres, as well as to the project development and management teams based at Beijing, Oxford and the 10 regional centres. The members of the steering committee and collaborative group are listed in the supplementary file. We thank Dr Yuanjie Pang for her help in revising this article.

**Collaborators** China Kadoorie Biobank Collaborative Group (see details in the online supplementary file).

**Contributors** JL and LL conceived and designed the paper. LL, ZC and JC, as members of the CKB steering committee, designed and supervised the conduct of the whole study, obtained funding and together with YG, ZB, LY, HD, YC, SYan and YZ acquired the data. SYang, MF, CY and JL analysed the data. SYang and

JL drafted the manuscript. CY, YG, ZB, MF, LY, HD, YC, SYan, YZ, JC, ZC, JL and LL contributed to the interpretation of the results and critical revision of the manuscript for important intellectual content. All authors contributed to and approved the final manuscript. JL is the study guarantor.

**Funding** This work was supported by grants (2016YFC0900500, 2016YFC0900501, 2016YFC0900504) from the National Key R&D Program of China. The CKB baseline survey and the first re-survey were supported by a grant from the Kadoorie Charitable Foundation in Hong Kong. The long-term follow-up is supported by grants from the UK Wellcome Trust (202922/Z/16/Z, 088158/Z/09/Z, 104085/Z/14/Z), National Natural Science Foundation of China (81390540, 81390541, 81390544) and Chinese Ministry of Science and Technology (2011BAI09B01). The funders had no role in the study design, data collection, data analysis and interpretation, writing of the report or the decision to submit the article for publication.

**Competing interests** None declared.

**Patient consent for publication** Not required.

**Ethics approval** The Ethical Review Committee of the Chinese Center for Disease Control and Prevention (Beijing, China) (Ethics ID number of approval: 005/2004) and the Oxford Tropical Research Ethics Committee, University of Oxford (UK) (Ethics ID number of approval: OXTREC 025-04).

**Provenance and peer review** Not commissioned; externally peer reviewed.

**Data availability statement** The dataset supporting the conclusions of this article is available from the study website (http://www.ckbiobank.org), along with the access policy and procedures.

**ORCID iD**
Jun Lv http://orcid.org/0000-0001-7916-3870

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
