## [Reviewer comments · BMJ Open]

ARTICLE DETAILS

TITLE (PROVISIONAL)	Bowel movement frequency and risks of major vascular and nonvascular diseases: a population-based cohort study among Chinese adults
AUTHORS	Yang, Songchun; Yu, Canqing; Guo, Yu; Bian, Zheng; Fan, Mengyu; Yang, Ling; Du, Huaidong; Chen, Yiping; Yan, Shichun; Zang, Yajing; Chen, Junshi; Chen, Zhengming; Lv, Jun; Li, Li-ming

VERSION 1 – REVIEW

REVIEWER	Shaw Watanabe Lifescience Promoting Association, Japan
REVIEW RETURNED	01-May-2019

GENERAL COMMENTS	This is a big work! The back story of this population seems that large amount of eating leads increase of stool as well as body weight gain. This obese trend is a cause of CVD, DM and hypertension. Waist circumference shows trend by bowel movement. Limitation of age (50-60 in average) seems to underestimate the incidence of diseases. Average age of "More than once a day group" is 3 years older than others, and it could influence the incidence rate. In Supplementary Table 3, significantly increased ORs are only present in old age groups among the less than 3 times a week group. Dietary habits were not dealt with precisely. Intake of dietary fiber links to the amount of feces with different microbiota. In the method section, the authors described about rice and other staple food in the questionnaire, so it is better to describe it if possible.
---

REVIEWER	Keiichi Sumida University of Tennessee Health Science Center, Division of Nephrology, Department of Medicine
REVIEW RETURNED	19-Aug-2019

GENERAL COMMENTS	In a large prospective cohort of 0.5 million adults aged 30 to 79 years from 10 diverse areas across China, Yang, et al.
--

	examined the association of bowel movement frequency assessed by self-reported questionnaire with risk of multiple vascular and nonvascular diseases over a median of 10 years follow-up. They found that both higher and lower frequencies of bowel movement were associated with higher risk of multiple vascular and nonvascular diseases. Given the lack of epidemiological evidence on the association of bowel movement and clinical outcomes, this study may be of clinical importance with a strength of its prospective study design with large sample size. However, several limitations and concerns still remain as detailed below. Minor comments:  1) One of the major concern is the effect of unmeasured confounders (e.g., laboratory data, medications, and comorbidities like cancer and neurodegenerative disease) on the outcomes. The authors should at least make clear why they were unable to obtain any additional information from laboratory data useful for risk indices (e.g., lipids) or pharmacy data (e.g., exposure to high risk drugs like opioids). 2) The potential pathophysiologic mechanisms underlying the association of bowel movement frequency with each outcome would be different between outcomes. This needs to be properly accounted for in the statistical analyses. 3) Although the authors mentioned the possibility of misclassification, using a self-reported history of diseases to ascertain baseline comorbidities can lead to a substantial misclassification of prevalent diseases. Other measures (e.g., prescription history) should thus be considered to ascertain baseline comorbidities. 4) Please explain how the authors treat competing events in their analyses. Minor comments:  1) In the Abstract, the study design should be specified. 2) Censoring events should be specified. 3) In the Methods, more detailed information on potential confounders should be provided (e.g., medications for cardiovascular diseases) 4) The prevalence of COPD and CKD should be added in Table 1. 5) Consider citing the following two articles related to constipation: J Am Soc Nephrol. 2017 Apr;28(4):1248-1258. and Atherosclerosis. 2019 Feb;281:114-120.
--	--

REVIEWER	Professor Alphons J.M. VERMORKEN Northwest University, College of Life Sciences
REVIEW RETURNED	01-Sep-2019

GENERAL COMMENTS	The paper: "Bowel movement frequency and risks of major vascular and nonvascular diseases: population based cohort study among 0.5 million Chinese adults", is a result of the China Kadoorie Biobank study that recruited half a million individuals between 2004 and 2008. During the 1990s, another large prospective cohort study had preceded the Kadoorie study in an attempt to determine the main avoidable causes of chronic disease in China. This latter study did not
--

foresee the collection of blood samples and consequently the range of possible risk factors that could be investigated was relatively limited. Therefore, the Kadoorie study involved the collection and long-term storage of blood samples. The Kadoorie study is among the largest blood-based prospective studies ever conducted in the world. Although its methodology appears well described and its results therefore potentially reproducible such a study is not easily reproduced for reasons of availability of funding and especially time. In order for the work to benefit the population it is therefore crucial that great efforts are done to assure that the results have the intended impact on the quality of public health and of health care in China and beyond, without delay! In this context it is important to note that the Kadoorie study anticipated that: "when sufficient numbers of people have developed or died from particular diseases, their stored blood will be retrieved and compared with blood retrieved from otherwise similar individuals who do not have this disease." (1) (reference 22).

In recent years the study has led to more than 150 publications about small aspects of the study and despite the undoubtedly good intentions of the authors and the Kadoorie study group, this implies a risk for fragmentation. As a consequence such publications may fail to have the intended impact on public health and health care. In order to avoid counterproductive fragmentation it is imperative to, at least, refer to related papers in the discussion of the results. This has regrettably not happened in the present paper. There is no reference to an earlier paper, by predominantly the same authors, describing the association between the frequency of bowel movements and the risk of colorectal cancer (2). This latter study, published only in the Chinese language, draws an interesting conclusion: "Participants who had a bowel movement frequency of more than once a day, had an increased risk of colorectal cancer in the subsequent five years. Since an abnormal increase of bowel movements is easily recognizable, programs should be set up on health self-management and early screening for colorectal cancer." This result could have a significant impact on public health provided such programs would be cost-effective. However, it is not sure that screening of individuals for colorectal cancer on the basis of increased bowel movement frequency alone would be cost-effective. This means that the cost of finding a case of colorectal cancer and of subsequent treatment of confirmed disease should be balanced in the context of health expenditure as a whole.

In the present paper it has been shown that the incidence of heart failure, a disease equally lethal than cancer, is increased in the category with more than one bowel movement daily. Ischaemic heart disease is also increased. This is the same category where increased risk for colorectal cancer has been noted. Recent and less recent literature shows that cancer and heart disease have many shared modifiable risk factors and therefore often even coexist in the same individuals (3, 4). Synergistic opportunities for screening may therefore increase cost-effectiveness (5). This demonstrates that discussing the results together increases their value.

	Inhabitual bowel habits may lead to metabolic abnormalities that may ultimately lead to disease. Alternatively individuals, with the lowest and highest bowel movement frequencies, can have these symptoms as a result of specific lifestyles, use of medication (including Chinese traditional medication), (sub)-clinical pathologies or genetic factors. In this context the authors note the lack of knowledge of the use of laxatives as a weakness of the study. This may be correct but the conclusion that the use of laxatives would probably not influence the outcome of the study may be wrong. While in the Chinese population laxative use is claimed to be limited to 5% of the population, it should be noted that in the present study the group with the lowest bowel movement frequency is even smaller than 5 %. Future studies would therefore most probably benefit from knowledge of the frequency and the type of laxatives, fiber or no-fiber based, used (6). Bowel movement frequency was associated with the risk of multiple vascular and non-vascular diseases, independent of traditional lifestyle factors for common non-communicable diseases, in the present study. This conclusion provokes questions and merits discussion. Consumption of fresh fruit has been shown to significantly lower blood pressure and blood glucose levels and to lower mortality from several major vascular and non-vascular diseases in the context of two Kadoorie studies, one of which was published in the New England Journal of Medicine (7, 8). The studies conclude: "Given the current low population level of fruit consumption, substantial health benefits could be gained from increased fruit consumption in China". Gelling fibers in fruit have been claimed to have a normalising effect on both stool frequency and consistency. Increased fresh fruit consumption will therefore also influence bowel movement frequency. Is it conceivable that fresh fruit consumption has a limited effect in the present study because it influences both the health outcomes and the transit time so that the relation between these latter two parameters is no longer significantly influenced by fruit consumption? Here again the authors have the opportunity to contribute to integration and to prevent fragmentation: - The authors should be encouraged to report on the effect of fresh fruit consumption on bowel movement frequency. The data must be available.- Another Kadoorie study reported on the differences in dietary intake frequency of adults in the ten participating Chinese regions (9). These studies may have confirmed the results of other studies claiming that fruit consumption is lower in the north of China (10, 11). As a result the nutritional status for folate is low and homocysteine levels and the prevalence of stroke are increased. Are bowel movement frequencies in regions in the north of China different from those in the south? The authors should be encouraged to investigate this. The data must be available. The authors claim:
--	--

	"Easily assessed bowel movement frequencies may be helpful to identify a change from usual pattern to abnormal bowel habits, one of the health signals that deserve attention". This has been claimed before, also by others, but it is not clear how the present study can lead to a conclusion about changes of bowel habits. Bowel movement frequency was self-reported only once at baseline. Studying the bowel movement frequencies in the north of China and in the south of China in relation to fruit consumption might demonstrate whether or not insufficient fruit consumption is one factor to look at when bowel movement frequency is inhabitual. As explained above individuals with the lowest and highest bowel movement frequencies form heterogeneous groups. Before determining the cost-effectiveness of screening for cancer and/or cardiovascular disease in such groups it would be better to identify subgroups by screening for easy to detect metabolic factors or for subclinical pathologies that play a role in both heart disease and (colorectal) cancer. One of the most important factors of strength of the Kadoorie study: the collection and long term storage of blood samples has not been exploited or even mentioned in the present manuscript! The authors discussed the possible role of oxidative stress as a causal factor associated with slow transit and coronary disease. Do the blood samples not allow for the verification of the role of oxidative stress by measuring for example MDA or probably even better oxidized-LDL? Other interesting opportunities are for example the measurement of folate, homocysteine and methylmalonic acid. Folate, unless taken as supplement can be an indicator for fruit consumption (12). Homocysteine is an indicator for nutrient deficiencies including the vitamins folate, B12, B6 and zinc. Methylmalonic acid is an indicator for vitamin B12 deficiency. The concentration of homocysteine in blood has been claimed to be higher in regions in northern China than southern China (13). This has been ascribed to a lower consumption of fruit (14). Both its concentration and its effects are also influenced by several gene polymorphisms (15). Hyperhomocysteinemia is claimed to be related to a higher incidence of hypertension and cardiovascular disease (14). Homocysteine also plays a role in both the initiation and progression of cancer (16, 17). Cardiac disease is associated with both subclinical hyper- and hypo-thyroid dysfunction although it is not yet clear what is the cause and what is the effect (18). Thyroid function is influencing oxidation of LDL (19, 20). Low and high bowel movement frequencies are associated with subclinical hypo- and hyperthyroidism respectively. The authors state: "Despite several biologically plausible mechanisms underlying the found associations have been suggested, further research is warranted." This sentence could be meaningful in a small study in which the authors have no possibility to go beyond the work performed. It is much less meaningful in a study that claims the collection and long term storage of blood samples as one of its major factors of strength. Conclusions:
--	--

This paper has results that merit to be published earlier or later. On the other hand in its present form its impact suffers significantly from fragmentation as described above. The Kadoorie studie should have much more to offer. There is also no good reason to delay a more integrated reporting, on the contrary: The consumption pattern and other lifestyle habits are changing rapidly in China. We are now seeing results based on data from individuals recruited before 2008. How long will these still be reflecting today's situation?

A minimum modification should include:

The paper on the association of bowel movement frequency with colorectal cancer needs to be referred to, its results need to be included in the form of tables and the results related to both cancer and the other diseases should be discussed in coherence.

The authors should refer to the earlier papers on the beneficial health effect of fruit consumption and to the paper on differences in dietary intake per region. They should do efforts, using the available data, to verify whether consumption of fresh fruit and whether consumption patterns in the southern regions of China lead to different bowel movement frequencies as compared to the northern regions where consumption of fresh fruit is known to be lower. If indeed fibers in fruit lead to normalisation of bowel movement frequencies this might be measurable.

Weaknesses of the study include a lack of knowledge of the use of laxatives and laxative types, of the use of medication and of traditional Chinese medication. Other factors that could improve a future study is registration of stool quality and repetition of the registration of bowel movement frequency five years after the start of the study. Individuals with non-satisfactory bowel movements may have taken action to improve their quality of life. Developing pathologies may have caused changes in bowel habits.

An explanation for not using the blood samples and therefore not (yet) exploiting the foreseen fantastic opportunities offered in the context of the Kadoorie study, as described above, is a minimum requirement for an acceptable manuscript.

A major modification of the paper:

using the stored blood samples for determining blood parameters such as for example oxidized LDL, folate, homocysteine, methylmalonic acid, thyroid hormones and some gene polymorphisms could bring the paper much nearer to concrete proposals for identifying causes of avoidable disease and death and thus for improving public health and health care. This could have a significant if not a dramatical influence on the paper's importance.

The Chinese population and indeed citizens worldwide might benefit if Health Authorities in China and elsewhere could encourage scientists and teams of scientists to do efforts for preventing fragmentation and for stimulating integration of data use in their reports. This should be done across the, in itself, valuable borders of medical specialties and scientific disciplines. Budgetary measures and structures for

recognition of such efforts by individual scientists and by teams of them, both at the National and at the International level, would be invaluable. Authorship of publications and impact factors of journals may not always pave the way to intrinsic motivation for achieving the goal pursued.

References:

- 1) Chen Z, Lee L, Chen J, Collins R, Wu F, Guo Y, Linksted P, Peto R.
Cohort profile: the Kadoorie Study of Chronic Disease in China (KSCDC).
Int J Epidemiol. 2005 Dec;34(6):1243-9. Epub 2005 Aug 30.
PMID: 16131516
- 2) Yang SC, Shen ZW, Yu CQ, Guo Y, Bian Z, Tan YL, Pei P, Wei YY, Chen F, Chen JS, Chen ZM, Lyu J, Li LM; China Kadoorie Biobank Collaborative Group.
[Association between the frequency of bowel movements and the risk of colorectal cancer in Chinese adults].
Zhonghua Liu Xing Bing Xue Za Zhi. 2019 Apr 10;40(4):382-388. doi: 10.3760/cma.j.issn.0254-6450.2019.04.003.
Chinese.
PMID: 31006195
- 3) Meijers WC, de Boer RA.
Common risk factors for heart failure and cancer.
Cardiovasc Res. 2019 Apr 15;115(5):844-853. doi: 10.1093/cvr/cvz035.
PMID: 30715247
- 4) Koene RJ, Prizment AE, Blaes A, Konety SH. Shared Risk Factors in Cardiovascular Disease and Cancer.
Circulation. 2016 Mar 15;133(11):1104-14. doi: 10.1161/CIRCULATIONAHA.115.020406. Review.
PMID: 26976915
- 5) Handy CE, Quispe R, Pinto X, Blaha MJ, Blumenthal RS, Michos ED, Lima JAC, Guallar E, Ryu S, Cho J, Kaye JA, Comin-Colet J, Corbella X, Cainzos-Achirica M. Synergistic Opportunities in the Interplay Between Cancer Screening and Cardiovascular Disease Risk Assessment.
Circulation. 2018 Aug 14;138(7):727-734. doi: 10.1161/CIRCULATIONAHA.118.035516.
PMID: 30359131
- 6) Citronberg JS, Hardikar S, Phipps A, Figueiredo JC, Newcomb P.
Laxative type in relation to colorectal cancer risk.
Ann Epidemiol. 2018 Oct;28(10):739-741. doi: 10.1016/j.annepidem.2018.06.011. Epub 2018 Jul 9.
PMID: 30150160
- 7) Du H, Li L, Bennett D, Guo Y, Key TJ, Bian Z, Sherliker P, Gao H, Chen Y, Yang L, Chen J, Wang S, Du R, Su H, Collins R, Peto R, Chen Z; China Kadoorie Biobank Study.
Fresh Fruit Consumption and Major Cardiovascular Disease in China.
N Engl J Med. 2016 Apr 7;374(14):1332-43. doi: 10.1056/NEJMoa1501451.
PMID: 27050205
- 8) Du H, Li L, Bennett D, Yang L, Guo Y, Key TJ, Bian Z, Chen Y, Walters RG, Millwood IY, Chen J, Wang J, Zhou X,

	Fang L, Li Y, Li X, Collins R, Peto R, Chen Z; China Kadoorie Biobank study. Fresh fruit consumption and all-cause and cause-specific mortality: findings from the China Kadoorie Biobank. Int J Epidemiol. 2017 Oct 1;46(5):1444-1455. doi: 10.1093/ije/dyx042. PMID: 28449053 9) Qin C, Yu C, Du H, Guo Y, Bian Z, Lyu J, Zhou H, Tan Y, Chen J, Chen Z, Li L. [Differences in diet intake frequency of adults: findings from half a million people in 10 areas in China]. Zhonghua Liu Xing Bing Xue Za Zhi. 2015 Sep;36(9):911-6. Chinese. PMID: 26814852 10) Zhang W, Li Y, Wang TD, Meng HX, Min GW, Fang YL, Niu XY, Ma LS, Guo JH, Zhang J, Sun MZ, Li CX. Nutritional status of the elderly in rural North China: a cross-sectional study. J Nutr Health Aging. 2014;18(8):730-6. doi: 10.1007/s12603-014-0034-2. PMID: 25286452 11) Dang S, Yan H, Zeng L, Wang Q, Li Q, Xiao S, Fan X. The status of vitamin B12 and folate among Chinese women: a population-based cross-sectional study in northwest China. PLoS One. 2014 Nov 12;9(11):e112586. doi: 10.1371/journal.pone.0112586. eCollection 2014. PMID: 25390898 12) Brevik A, Vollset SE, Tell GS, Refsum H, Ueland PM, Loeken EB, Drevon CA, Andersen LF. Plasma concentration of folate as a biomarker for the intake of fruit and vegetables: the Hordaland Homocysteine Study. Am J Clin Nutr. 2005 Feb;81(2):434-9. PMID: 15699232 13) Yang B, Fan S, Zhi X, Wang Y, Wang Y, Zheng Q, Sun G. Prevalence of hyperhomocysteinemia in China: a systematic review and meta-analysis. Nutrients. 2014 Dec 29;7(1):74-90. doi: 10.3390/nu7010074. Review. PMID: 25551247 14) Yang B, Fan S, Zhi X, He J, Ma P, Yu L, Zheng Q, Sun G. Interactions of homocysteine and conventional predisposing factors on hypertension in Chinese adults. J Clin Hypertens (Greenwich). 2017 Nov;19(11):1162-1170. doi: 10.1111/jch.13075. PMID: 28942612 15) Li WX, Lv WW, Dai SX, Pan ML, Huang JF. Joint associations of folate, homocysteine and MTHFR, MTR and MTRR gene polymorphisms with dyslipidemia in a Chinese hypertensive population: a cross-sectional study. Lipids Health Dis. 2015 Sep 4;14:101. doi: 10.1186/s12944-015-0099-x. PMID: 26337056 16) Hasan T, Arora R, Bansal AK, Bhattacharya R, Sharma GS, Singh LR. Disturbed homocysteine metabolism is associated with cancer. Exp Mol Med. 2019 Feb 21;51(2):21. doi: 10.1038/s12276-019-0216-4. Review. PMID: 30804341
--	--

	17) Liu Z, Cui C, Wang X, Fernandez-Escobar A, Wu Q, Xu K, Mao J, Jin M, Wang K. Plasma Levels of Homocysteine and the Occurrence and Progression of Rectal Cancer. Med Sci Monit. 2018 Mar 27 ;24:1776-1783. doi: 10.12659/MSM.909217. PMID: 29581416 18) Razvi S, Jabbar A, Pingitore A, Danzi S, Biondi B, Klein I, Peeters R, Zaman A, Iervasi G. Thyroid Hormones and Cardiovascular Function and Diseases. J Am Coll Cardiol. 2018 Apr 24;71(16):1781-1796. doi: 10.1016/j.jacc.2018.02.045. Review. PMID: 29673469 19) Oge A, Sozmen E, Karaoglu AO. Effect of thyroid function on LDL oxidation in hypothyroidism and hyperthyroidism. Endocr Res. 2004 Aug;30(3):481-9. PMID: 15554363 20) Oktay S, Uslu L, Emekli N. Effects of altered thyroid states on oxidative stress parameters in rats. J Basic Clin Physiol Pharmacol. 2017 Mar 1;28(2):159-165. doi: 10.1515/jbcpp-2015-0113. PMID: 27824612
--	---

VERSION 1 – AUTHOR RESPONSE

Reviewer 1: Dr. Shaw Watanabe

Comment 1: This is a big work!

Response: We thank Dr. Watanabe for this encouraging comment.

Comment 2: The back story of this population seems that large amount of eating leads increase of stool as well as body weight gain. This obese trend is a cause of CVD, DM and hypertension. Waist circumference shows trend by bowel movement.

Response: We thank Dr. Watanabe for providing a potential explanation for what we have observed in this cohort study. Considering potential confounding by excessive caloric intake, we adjusted for BMI (kg/m²) and waist circumference (cm) in our analyses (Table 2). After these adjustments, more frequent bowel movements were still associated with T2DM, IHD, and heart failure, indicating other possible mechanism underlying these associations.

Comment 3: Limitation of age (50-60 in average) seems to underestimate the incidence of diseases. Average age of "More than once a day group" is 3 years older than others, and it

could influence the incidence rate. In Supplementary Table 3, significantly increased ORs are only present in old age groups among the less than 3 times a week group.

Response: The age of CKB participants ranged from 30 to 79 years, and the mean age was 51.5 years. In general, the age range of the study population may limit the extrapolability of the results, but it is less likely to harm the internal validity in the association study.

We agree with Dr. Watanabe that there were some differences in the average ages among different groups of participants. As mentioned in the methods section, we used Cox proportional hazards regression model with age as the underlying time scale and stratified by 5-year groups and 10 survey sites. After further adjustment age in the model, the results almost remained unchanged. Such measures may help control for confounding by age.

Supplementary Table 3 shows whether the associations of BMF with multiple vascular and nonvascular diseases were consistent in different age groups, i.e., the effect modification by age on the association. The HRs of IHD and MCEs among “the less than 3 times a week” group in old age groups were only present in old age group (Supplementary Table 3). However, their *P* values for interaction were not significant after a Bonferroni-correction ($\alpha=0.00625$). The results indicate that the associations of BMF with multiple health outcomes were consistent across different age groups.

Comment 4: Dietary habits were not dealt with precisely. Intake of dietary fiber links to the amount of feces with different microbiota. In the method section, the authors described about rice and other staple food in the questionnaire, so it is better to describe it if possible.

Response: We thank Dr. Watanabe for pointing out this limitation. We acknowledge that dietary frequency questionnaire used in our study was not comprehensive. We only asked participants about the five categories of frequency (daily, 4 to 6 days per week, 1 to 3 days per week, monthly, or never or rarely) of 12 major food groups during the previous 12 months at baseline survey. Unfortunately, the amounts of consumption were not collected.

We have included the intake frequencies of rice, wheat, and other staple food in the sensitivity analyses. The results almost remained unchanged after adjusting for these dietary factors. We have added the qualitative food frequency questionnaire in the supplementary file and provided a brief explanation about the questionnaire in the method section “Assessment of covariates” (page 7: line 6 - 8).

Comment 5: (Please see the attached paper.)

Response: We have noticed that there was a paper provided by Dr. Watanabe as the attachment². Our understanding is that Dr. Watanabe wanted to know what was the reference for the cut-off point of waist circumference used in our manuscript. We have added a reference in the text (page 10: line 2). We have also tried to use the cut-off point for Japanese and that recommended by the International Diabetes Foundation (IDF) in the analyses. The results did not change appreciably (data not shown). Therefore, we maintained our results in Table 3.

Reviewer 2: Dr. Keiichi Sumida

Comment 1: In a large prospective cohort of 0.5 million adults aged 30 to 79 years from 10 diverse areas across China, Yang, et al. examined the association of bowel movement frequency assessed by self-reported questionnaire with risk of multiple vascular and nonvascular diseases over a median of 10 years follow-up. They found that both higher and lower frequencies of bowel movement were associated with higher risk of multiple vascular and nonvascular diseases. Given the lack of epidemiological evidence on the association of bowel movement and clinical outcomes, this study may be of clinical importance with a strength of its prospective study design with large sample size. However, several limitations and concerns still remain as detailed below.

Response: We thank Dr. Sumida for your encouraging comment. Detailed replies and where we have revised our manuscript are shown below.

Major comments:

Comment 2: One of the major concern is the effect of unmeasured confounders (e.g., laboratory data, medications, and comorbidities like cancer and neurodegenerative disease) on the outcomes. The authors should at least make clear why they were unable to obtain any additional information from laboratory data useful for risk indices (e.g., lipids) or pharmacy data (e.g., exposure to high risk drugs like opioids).

Response: In the China Kadoorie Biobank cohort, we did collect a 10-ml non-fasting blood sample for each participant at baseline. However, the research team does not have enough money to test all blood samples up till now.

As for medication data, we have collected information about consumption of nutritional supplements (e.g., fish oil/cod liver oil, vitamins, calcium/iron/zinc, ginseng and other herbal products), and medications for cardiovascular diseases (e.g. aspirin, ACE-I,

beta-blocker, statins, diuretics, and Ca²⁺ antagonist). Due to the limited time of the investigation, we did not collect more information about other medication use at baseline survey, such as opioids use. We have added this limitation in the discussion section of the revised manuscript (page 17: line 13 - 14).

We excluded the participants who reported to have been diagnosed as cancer at baseline in our analyses. Information about common neurodegenerative diseases including Alzheimer's disease and Parkinson's disease were not collected at baseline. However, individuals who had a known diagnosed disease like Alzheimer's disease were also less likely to participate in our study.

Comment 3: The potential pathophysiologic mechanisms underlying the association of bowel movement frequency with each outcome would be different between outcomes. This needs to be properly accounted for in the statistical analyses.

Response: We agree with Dr. Sumida that different statistical analysis strategies are required for examining the associations of BMF with different health outcomes. In our analyses, the difference in the analysis strategies reflected in the inclusion criteria of participants and the potential confounders adjusted in the regression models.

In the present study, participants with disease of interest at baseline were further excluded when conducting analyses of particular disease outcome (page 6: line 2 - 10). For example, when we considered COPD as the health outcome, we further excluded participants who had a self-reported history of emphysema, bronchitis or pulmonary heart disease, and who was spirometry-measured to have airflow obstruction (AFO) at baseline.

Also, we have adjusted for the potential confounders respectively for each disease outcome in the Cox regression models. For example, the family history of certain disease was only adjusted for corresponding disease analysis, as long as we have this information. Prevalent hypertension and diabetes at baseline were adjusted for cardiovascular diseases while not adjusted for T2DM. We also considered other potential confounding factors in the sensitivity analyses, such as medications of cardiovascular diseases for IHD and prevalent asthma for COPD.

Comment 4: Although the authors mentioned the possibility of misclassification, using a self-reported history of diseases to ascertain baseline comorbidities can lead to a substantial misclassification of prevalent diseases. Other measures (e.g., prescription history) should thus be considered to ascertain baseline comorbidities.

Response: In the present study, participants were asked to report whether they had ever been diagnosed with a disease by a doctor, the age of their first diagnosis, and whether they were still on treatment at baseline survey. As for diabetes and COPD, we measured blood glucose and lung function at baseline survey in addition to self-reported history of these diseases. Hypertension in this study was defined as those who reported a diagnosis of hypertension by a physician, use of antihypertensives at baseline, or measured to be hypertensive at baseline.³ We have recognized that, if we relied solely on self-reported information, there might be a small number of cases who were on treatment while they were not very clear about the diagnosis of the diseases. These cases would be misclassified as healthy individuals. In the sensitivity analysis, we have excluded cases with records in local disease registries, death registries or national health insurance system during the first two years, which to some extent help minimize the influence of the missing report of prevalent cases at baseline. The results almost remained unchanged.

Comment 5: Please explain how the authors treat competing events in their analyses.

Response: Competing risks occur when there are at least two possible ways that a person can fail, but only one such failure type can actually occur.⁴ In the present analyses, if more than one health outcomes occurred in the same participant, we would regard this participant as a case for every health outcome that has occurred. Therefore, we did not perform the competing risks survival analysis. There are several previous studies that are similar to ours following same statistical strategy.⁵⁻⁸

Minor comments:

Comment 6: In the Abstract, the study design should be specified.

Response: We have added study design to the abstract of the revised manuscript (page 2: line 3 - 4).

Comment 7: Censoring events should be specified.

Response: In the “statistical analysis” of method section, we described that other causes of death, loss to follow-up, or global censoring date (31 December 2016) were the censoring events of our study (page 8: line 17 - 18).

Comment 8: In the Methods, more detailed information on potential confounders should be provided (e.g., medications for cardiovascular diseases)

Response: We have listed the covariates that were regarded as potential confounders in the “statistical analysis” of method section (page 9: line 2 - 8). Details about medications for cardiovascular diseases and dietary supplements are provided (page 9: line 14 - 17). Due to the word count limit, the categories for each covariate were exhibited at the footnotes for Table 2 and the supplementary file. If editors and reviewers think it is better to include them in the text, we would also be pleased to do that.

Comment 9: The prevalence of COPD and CKD should be added in Table 1.

Response: As mentioned in the “Subject and Methods – Study population” section, we excluded participants who had a self-reported history of emphysema, bronchitis or pulmonary heart disease or who were spirometry-measured to have airflow obstruction (AFO) at baseline, which was defined as prevalent COPD at baseline, for the analysis of COPD. Also, we excluded participants who had a self-reported history of kidney disease for the analysis of CKD (page 6: line 4 - 7, 9 - 10). Since prevalent COPD and kidney disease were not regarded as confounders in the analyses of other health outcomes, we did not report them in Table 1.

Comment 10: Consider citing the following two articles related to constipation: J Am Soc Nephrol. 2017 Apr;28(4):1248-1258. and Atherosclerosis. 2019 Feb;281:114-120.

Response: We thank Dr. Sumida for reminding us to find out all related studies of this field in time. The first one published in 2017 is about the association of constipation and incident CKD.⁹ The latter one published in 2019 is about the association of constipation and risk of death and cardiovascular events.¹⁰ We have cited both of them and revised our discussion in the manuscript (page 13: line 18 - 21, page 14: 1 - 3; page 15: line 3 - 7).

Reviewer 3: Prof. Alphons J.M. Vermorcken

General comments:

Paragraph 1: The paper: "Bowel movement frequency and risks of major vascular and nonvascular diseases: population based cohort study among 0.5 million Chinese adults", is a result of the China Kadoorie Biobank study that recruited half a million individuals between 2004 and 2008. During the 1990s, another large prospective cohort study had preceded the Kadoorie study in an attempt to determine the main avoidable causes of chronic disease in China. This latter study did not foresee the collection of blood samples and consequently the range of possible risk factors that could be investigated was relatively limited. Therefore, the Kadoorie study involved the collection and long-term storage of blood samples. The Kadoorie study is among the largest blood-based prospective studies ever conducted in the world. Although its methodology appears well described and its results therefore potentially reproducible such a study is not easily reproduced for reasons of availability of funding and especially time. In order for the work to benefit the population it is therefore crucial that great efforts are done to assure that the results have the intended impact on the quality of public health and of health care in China and beyond, without delay! In this context it is important to note that the Kadoorie study anticipated that: "when sufficient numbers of people have developed or died from particular diseases, their stored blood will be retrieved and compared with blood retrieved from otherwise similar individuals who do not have this disease." (1) (reference 22).

Paragraph 2: In recent years the study has led to more than 150 publications about small aspects of the study and despite the undoubtedly good intentions of the authors and the Kadoorie study group, this implies a risk for fragmentation. As a consequence such publications may fail to have the intended impact on public health and health care. In order to avoid counterproductive fragmentation it is imperative to, at least, refer to related papers in the discussion of the results. This has regrettably not happened in the present paper. There is no reference to an earlier paper, by predominantly the same authors, describing the association between the frequency of bowel movements and the risk of colorectal cancer (2). This latter study, published only in the Chinese language, draws an interesting conclusion: "Participants who had a bowel movement frequency of more than once a day, had an increased risk of colorectal cancer in the subsequent five years. Since an abnormal increase of bowel movements is easily recognizable, programs should be set up on health self-management and early screening for colorectal cancer." This result could have a significant impact on public health provided such programs would be cost-effective. However, it is not sure that screening of individuals for colorectal cancer on the basis of increased bowel movement frequency alone would be cost-effective. This means that the cost of finding a case of colorectal cancer and of subsequent treatment of confirmed disease should be balanced in the context of health expenditure as a whole.

Paragraph 3: In the present paper it has been shown that the incidence of heart failure, a disease equally lethal than cancer, is increased in the category with more than one bowel movement daily. Ischaemic heart disease is also increased. This is the same category where increased risk for colorectal cancer has been noted. Recent and less recent literature shows that cancer and heart disease have many shared modifiable risk factors and therefore often even coexist in the same individuals (3, 4). Synergistic opportunities for screening may therefore increase cost-effectiveness (5). This demonstrates that discussing the results together increases their value.

Paragraph 4: Inhabital bowel habits may lead to metabolic abnormalities that may ultimately lead to disease. Alternatively individuals, with the lowest and highest bowel movement frequencies, can have these symptoms as a result of specific lifestyles, use of medication (including Chinese traditional medication), (sub)-clinical pathologies or genetic factors. In this context the authors note the lack of knowledge of the use of laxatives as a weakness of the study. This may be correct but the conclusion that the use of laxatives would probably not influence the outcome of the study may be wrong. While in the Chinese population laxative use is claimed to be limited to 5% of the population, it should be noted that in the present study the group with the lowest bowel movement frequency is even smaller than 5%. Future studies would therefore most probably benefit from knowledge of the frequency and the type of laxatives, fiber or no-fiber based, used (6).

Paragraph 5: Bowel movement frequency was associated with the risk of multiple vascular and non-vascular diseases, independent of traditional lifestyle factors for common non-communicable diseases, in the present study. This conclusion provokes questions and merits discussion. Consumption of fresh fruit has been shown to significantly lower blood pressure and blood glucose levels and to lower mortality from several major vascular and non-vascular diseases in the context of two Kadoorie studies, one of which was published in the New England Journal of Medicine (7, 8). The studies conclude: "Given the current low population level of fruit consumption, substantial health benefits could be gained from increased fruit consumption in China". Gelling fibers in fruit have been claimed to have a normalising effect on both stool frequency and consistency. Increased fresh fruit consumption will therefore also influence bowel movement frequency. Is it conceivable that fresh fruit consumption has a limited effect in the present study because it influences both the health outcomes and the transit time so that the relation between these latter two parameters is no longer significantly influenced by fruit consumption?

Paragraph 6: Here again the authors have the opportunity to contribute to integration and to prevent fragmentation:

- The authors should be encouraged to report on the effect of fresh fruit consumption on bowel movement frequency. The data must be available.

- Another Kadoorie study reported on the differences in dietary intake frequency of adults in the ten participating Chinese regions (9). These studies may have confirmed the results of other studies claiming that fruit consumption is lower in the north of China (10, 11). As a result the nutritional status for folate is low and homocysteine levels and the prevalence of stroke are increased. Are bowel movement frequencies in regions in the north of China different from those in the south? The authors should be encouraged to investigate this. The data must be available.

Paragraph 7: The authors claim: "Easily assessed bowel movement frequencies may be helpful to identify a change from usual pattern to abnormal bowel habits, one of the health signals that deserve attention". This has been claimed before, also by others, but it is not clear how the present study can lead to a conclusion about changes of bowel habits. Bowel movement frequency was self-reported only once at baseline. Studying the bowel movement frequencies in the north of China and in the south of China in relation to fruit consumption might demonstrate whether or not insufficient fruit consumption is one factor to look at when bowel movement frequency is in habitual.

Paragraph 8: As explained above individuals with the lowest and highest bowel movement frequencies form heterogeneous groups. Before determining the cost-effectiveness of screening for cancer and/or cardiovascular disease in such groups it would be better to identify subgroups by screening for easy to detect metabolic factors or for subclinical pathologies that play a role in both heart disease and (colorectal) cancer. One of the most important factors of strength of the Kadoorie study: the collection and long term storage of blood samples has not been exploited or even mentioned in the present manuscript!

Paragraph 9: The authors discussed the possible role of oxidative stress as a causal factor associated with slow transit and coronary disease. Do the blood samples not allow for the verification of the role of oxidative stress by measuring for example MDA or probably even better oxidized-LDL? Other interesting opportunities are for example the measurement of folate, homocysteine and methylmalonic acid. Folate, unless taken as supplement can be an indicator for fruit consumption (12). Homocysteine is an indicator for nutrient deficiencies including the vitamins folate, B12, B6 and zinc. Methylmalonic acid is an indicator for vitamin B12 deficiency. The concentration of homocysteine in blood has been claimed to be higher in regions in northern China than southern China (13). This has been ascribed to a lower consumption of fruit (14). Both its concentration and its effects are also influenced by several gene polymorphisms (15). Hyperhomocysteinemia is claimed to be related to a higher

incidence of hypertension and cardiovascular disease (14). Homocysteine also plays a role in both the initiation and progression of cancer (16, 17).

Paragraph 10: Cardiac disease is associated with both subclinical hyper- and hypo-thyroid dysfunction although it is not yet clear what is the cause and what is the effect (18). Thyroid function is influencing oxidation of LDL (19, 20). Low and high bowel movement frequencies are associated with subclinical hypo- and hyperthyroidism respectively. The authors state: "Despite several biologically plausible mechanisms underlying the found associations have been suggested, further research is warranted." This sentence could be meaningful in a small study in which the authors have no possibility to go beyond the work performed. It is much less meaningful in a study that claims the collection and long term storage of blood samples as one of its major factors of strength.

Paragraph 11: Conclusions: This paper has results that merit to be published earlier or later. On the other hand in its present form its impact suffers significantly from fragmentation as described above. The Kadoorie studie should have much more to offer. There is also no good reason to delay a more integrated reporting, on the contrary: The consumption pattern and other lifestyle habits are changing rapidly in China. We are now seeing results based on data from individuals recruited before 2008. How long will these still be reflecting today's situation?

Response: We thank Prof. Vermorken for reviewing our manuscript so carefully and providing us lots of important and insightful comments. In order to make our replies more clearly, we first summarized reviewer's comments above into the following nine points. Thereafter, we responded these points together with the comments following the titles of "minimum modification" and "major modification" one by one.

(NOTE: The reference in the original comments is omitted.)

The main comments summarized are as below:

Point 1: The Kadoorie study is among the largest blood-based prospective studies ever conducted in the world (from **Paragraph 1**). The collection and long term storage of blood samples has not been exploited or even mentioned in the present manuscript! (from **Paragraph 8**) Do the blood samples not allow for the verification of the role of oxidative stress by measuring, for example, MDA, oxidized-LDL, homocysteine (from **Paragraph 9**), or thyroid function (from **Paragraph 10**)? (*this point will be discussed together with **Comment 1** below*)

Point 2: In order to avoid counterproductive fragmentation, it is imperative to refer to related papers in the discussion of the results. In the present paper, there is no reference to an earlier paper describing the association between the frequency of bowel movements and the risk of colorectal cancer (from **Paragraph 2**). Recent literature shows that cancer and heart disease have many shared modifiable risk factors and therefore often even coexist in the same individuals. Discussing the results together will increase the value of this paper (from **Paragraph 3**). *(this point will be discussed together with **Comment 3** below)*

Point 3: The conclusion that the use of laxatives would probably not influence the outcome of the study may be wrong (from **Paragraph 4**). *(this point will be discussed together with **Comment 5** below)*

Point 4: Gelling fibers in the fruit have been claimed to have a normalising effect on both stool frequency and consistency. Increased fresh fruit consumption will therefore also influence bowel movement frequency. Is the relation between bowel movement frequency and health outcomes not significantly influenced by fruit consumption? (from **Paragraph 5**) The authors should be encouraged to report on the effect of fresh fruit consumption on bowel movement frequency (from **Paragraph 6**). *(this point will be discussed together with **Comment 4** below)*

Point 5: Are bowel movement frequencies in regions in the north of China different from those in the south? The authors should be encouraged to investigate this (from **Paragraph 6**). *(this point will be discussed together with **Comment 4** below)*

Point 6: It is not clear how the present study can lead to a conclusion about changes of bowel habits. Bowel movement frequency was self-reported only once at baseline (from **Paragraph 7**).

Response: We have deleted “Easily assessed BMF may be helpful to identify a change from usual pattern to abnormal bowel habits, one of the health signals that deserve attention” in our conclusion.

Point 7: Before determining the cost-effectiveness of screening for cancer and/or cardiovascular disease in abnormal bowel movement frequency groups, it would be better to identify subgroups by screening for easy to detect metabolic factors or for subclinical pathologies that play a role in both heart disease and (colorectal) cancer (from **Paragraph 8**). *(this point will be discussed together with **Comment 1** below)*

Point 8: The authors state: "Despite several biologically plausible mechanisms underlying the found associations have been suggested, further research is warranted." This sentence could be meaningful in a small study but less meaningful in a study that claims the collection and long term storage of blood samples as one of its major factors of strength. (from **Paragraph 10**) *(this point will be discussed together with **Comment 1** below)*

Point 9: The consumption pattern and other lifestyle habits are changing rapidly in China. We are now seeing results based on data from individuals recruited before 2008. How long will these still be reflecting today's situation? (from **Paragraph 11**)

Response: Participants of CKB have been prospectively followed up for more than 10 years from baseline recruitment to the end of 2016. Due to the huge workload and financial input, we are unable to survey every participant face to face regularly to catch their changing consumption pattern and other lifestyle habits. We acknowledge this limitation and have added this to the discussion section (page 17: line 10 - 11).

Here are our responses to the major and minor comments:

Major comments:

Comment 1: Using the stored blood samples for determining blood parameters such as for example oxidized LDL, folate, homocysteine, methylmalonic acid, thyroid hormones and some gene polymorphisms could bring the paper much nearer to concrete proposals for identifying causes of avoidable disease and death and thus for improving public health and health care. This could have a significant if not a dramatical influence on the paper's importance.

Response: We thank Prof. Vermorken for suggesting several good research ideas for future studies. In the China Kadoorie Biobank, we did collect a 10-ml non-fasting blood sample for

each participant. However, the research team does not have enough money to test all blood samples up till now. We have added this limitation to the discussion section (page 17: line 15).

Comment 2: The Chinese population and indeed citizens worldwide might benefit if Health Authorities in China and elsewhere could encourage scientists and teams of scientists to do efforts for preventing fragmentation and for stimulating integration of data use in their reports. This should be done across the, in itself, valuable borders of medical specialties and scientific disciplines. Budgetary measures and structures for recognition of such efforts by individual scientists and by teams of them, both at the National and at the International level, would be invaluable. Authorship of publications and impact factors of journals may not always pave the way to intrinsic motivation for achieving the goal pursued.

Response: This is exactly what we hope to see in the future, and we believe that our scientific community around the world will achieve this goal step by step.

Minor comments:

Comment 3: The paper on the association of bowel movement frequency with colorectal cancer needs to be referred to, its results need to be included in the form of tables and the results related to both cancer and the other diseases should be discussed in coherence.

Response: The paper of the association of BMF with colorectal cancer (CRC) has been published in *Chinese Journal of Epidemiology* in 2019.¹¹ In this study, participants who had BMF more than once a day appeared an increased risk of CRC in the subsequent five years, while such associations became statistically insignificant in the subsequent follow-up, indicating that abnormal increase of BMF may be an early symptom of CRC. We have added to cite this paper in the “Introduction” section (page 4: line 11 - 14).

However, in the present study, the associations of higher BMF with IHD, heart failure, COPD, T2DM and CKD remained significant even after we excluded cases occurred in the first five years of follow-up (data were not shown). We assumed that higher BMF is associated with the altered gut microbiota that contributes to the development and progression of diseases such as atherosclerosis, heart failure, CKD, T2DM, and COPD. Considering different underlying mechanisms and duplicate publication, we did not show the result of CRC in the present paper.

Comment 4: The authors should refer to the earlier papers on the beneficial health effect of fruit consumption and to the paper on differences in dietary intake per region. They should do efforts, using the available data, to verify whether consumption of fresh fruit and whether consumption patterns in the southern regions of China lead to different bowel movement frequencies as compared to the northern regions where consumption of fresh fruit is known to be lower. If indeed fibers in fruit lead to normalisation of bowel movement frequencies this might be measurable.

Response: Following the reviewer’s suggestion, we added analyses as below. The proportion for less frequent bowel movement seems to be higher among those with the lowest frequency of fresh fruit consumption (Graph 1). Participants in the north and south regions had a similar distribution of BMF (Graph 2).

However, the purpose of the present study was to investigate the associations of BMF with multiple health outcomes. We have shown the proportion of regular consumption (at least 4-6 days per week) of fresh fruits and proportion of urban residents among different BMF groups. We stratified our analyses in ten survey sites in the Cox regression models to account for the effect of different regions (page 8: line 19 - 21). We also adjusted the intake frequency of fresh fruits as a potential confounder and the results almost remained unchanged before and after adjustment (Supplementary Table 1), indicating that fresh fruits consumption had little confounding effect on the associations of BMF with multiple health outcomes.

Considering the following results are not directly relevant to the purpose of this study, we would tend not to add them to our revised manuscript. We have taken the suggestion of Prof. Vermorken to refer to the earlier papers that came from our previous studies in the revised manuscript (page 16: line 4 - 6).

Graph 1. Distribution of BMF according to different intake frequency groups of fresh fruit (n=487,198)

Note: Age (continuous), gender (male, female) and region (10 sites) were adjusted in the multinomial logistic regression model.

Graph 2. Distribution of BMF in north regions and south regions (n=487,198)

Note: Age (continuous), gender (male, female) were adjusted in the multinomial logistic regression model. North regions include Qingdao (U), Harbin (U), Gansu (R), Henan (R); South regions include Haikou (U), Suzhou (U), Liuzhou (U), Sichuan (R), Zhejiang (R), Hunan (R). “U” means “urban”, “R” means “rural”.

Comment 5: Weaknesses of the study include a lack of knowledge of the use of laxatives and laxative types, of the use of medication and of traditional Chinese medication. Other factors that could improve a future study is registration of stool quality and repetition of the registration of bowel movement frequency five years after the start of the study. Individuals with non-satisfactory bowel movements may have taken action to improve their quality of life. Developing pathologies may have caused changes in bowel habits.

Response: We acknowledge these limitations and have made some revisions in the discussion. The explanation of the prevalence of laxative use in China “However, the prevalence of laxative use was as low as 5% in the Chinese population, that may not cause a significant influence on the overall results” has been deleted (page 17: line 13).

If a part of individuals with non-satisfactory bowel movements had taken actions to improve their quality of life, we could infer that risks of multiple diseases might be a little lower than expected in this group, which would attenuate the observed associations.

Also, changes in bowel habits might reflect the developing pathologies in the gut. As we have discussed in the response to **Comment 3**, the associations of higher BMF with the health outcomes in this study remained statistically significant even after we excluded cases occurred in the first five years of follow-up (data not shown). This result may rule out the possibility of reverse causality to some extent.

Comment 6: An explanation for not using the blood samples and therefore not (yet) exploiting the foreseen fantastic opportunities offered in the context of the Kadoorie study, as described above, is a minimum requirement for an acceptable manuscript.

Response: Please see response to **Comment 1**.

References

1. Yu CQ, Shi ZM, Lv J, et al. Major Dietary Patterns in Relation to General and Central Obesity among Chinese Adults. *Nutrients* 2015;7(7):5834-49. doi:10.3390/nu7075253
2. Division of Disease Control, Ministry of Health of the People's Republic of China. Guideline for prevention and control of overweight and obesity in Chinese adults. Beijing: People's Medical Publishing House 2006.
3. Nakade M, Morita A, Watanabe S, et al. Rationale Diagnostic Criteria of the Metabolic Syndrome. *Diabetes Res Open J* 2019;4(1):18-24. doi:10.17140/DROJ-4-139
4. Lewington S, Lacey B, Clarke R, et al. The Burden of Hypertension and Associated Risk for Cardiovascular Mortality in China. *JAMA Intern Med* 2016;176(4):524-32. doi:10.1001/jamainternmed.2016.0190
5. Kleinbaum DG, Klein M. Survival Analysis: A Self-Learning Text, Third Edition. United States of America: Springer 2012:391-458.
6. Zoungas S, Patel A, Chalmers J, et al. Severe hypoglycemia and risks of vascular events and death. *N Engl J Med* 2010;363(15):1410-8. doi:10.1056/NEJMoa1003795
7. Millwood IY, Bennett DA, Holmes MV, et al. Association of CETP Gene Variants With Risk for Vascular and Nonvascular Diseases Among Chinese Adults. *JAMA Cardiol* 2018;3(1):34-43. doi:10.1001/jamacardio.2017.4177

8. Lacey B, Lewington S, Clarke R, et al. Age-specific association between blood pressure and vascular and non-vascular chronic diseases in 0.5 million adults in China: a prospective cohort study. *Lancet Glob Health* 2018;6(6):e641-e49. doi:10.1016/s2214-109x(18)30217-1
9. Galbete C, Kroger J, Jannasch F, et al. Nordic diet, Mediterranean diet, and the risk of chronic diseases: the EPIC-Potsdam study. *BMC Med* 2018;16(1):99. doi:10.1186/s12916-018-1082-y
10. Sumida K, Molnar MZ, Potukuchi PK, et al. Constipation and Incident CKD. *J Am Soc Nephrol* 2017;28(4):1248-58. doi:10.1681/ASN.2016060656
11. Sumida K, Molnar MZ, Potukuchi PK, et al. Constipation and risk of death and cardiovascular events. *Atherosclerosis* 2019;281:114-20. doi:10.1016/j.atherosclerosis.2018.12.021
12. Yang S, Shen Z, Yu C, et al. Association between the frequency of bowel movements and the risk of colorectal cancer in Chinese adults. *Chin J Epidemiol* 2019;40(4):382-88. doi:10.3760/cma.j.issn.0254-6450.2019.04.003

VERSION 2 – REVIEW

REVIEWER	Keiichi Sumida University of Tennessee Health Science Center, Division of Nephrology, Department of Medicine
REVIEW RETURNED	04-Oct-2019

GENERAL COMMENTS	The manuscript has been much improved after revision. Just one comment: Although the authors responded that kidney disease is not regarded as a confounder, given the association of CKD with bowel movement frequency (Toxins. May 16 2018;10(5)) and with outcomes, it should be recognized as a confounder.
--

REVIEWER	A.J.M. Vermorken Northwest University, College of Life Sciences
REVIEW RETURNED	18-Oct-2019

GENERAL COMMENTS	Review 2 The authors have done a rapid and efficient job in their efforts to take each remark of the reviewers into account. I am grateful for their work. Nevertheless it is not always possible to deal with an issue in the manuscript by adding a sentence in the original text only. A few issues therefore remain to be discussed.
---

The authors state that: "A properly functioning gastrointestinal tract plays an essential role in health". They argue that bowel movement frequency may represent a simple quantifiable indicator of adequate colonic function. This is correct and for this reason associations between bowel movement frequency and incidence of disease might be used for the purpose of prevention and early detection of disease. Providing scientific evidence for prevention and control of non-communicable diseases has been declared to be the goal of the Kadoorie Studies as published previously by a team including some of the present authors (1). Nevertheless the authors fail to describe how their data can contribute to their own goal.

1) BMJ Open states that it focusses on research that is relevant to patients and clinicians. The present paper (still) fails to spell out its relevance for patients and clinicians. The authors state in their response letter that: "the purpose of the present study was to investigate the associations of bowel movement frequency with multiple health outcomes". That is the purpose for the epidemiologist. The relevance of these associations for patients and clinicians have, however, to be discussed in the relevant context in which previous papers of the same authors also play a role. While in the first manuscript a not completely correct effort was made to describe the relevance in the "Conclusions" this paragraph has now been modified without providing a "message" for patients or clinicians. Moreover, it is unclear how the data obtained can lead to the second part of the new statement: "Our findings highlight that gastrointestinal health is related to the health of multiple systems and provide preliminary epidemiological evidence for potential systemic effects of gut microbiota". The microbiota were discussed as a relevant factor and others have shown that it plays a role but do the results of the present paper provide evidence for the involvement of the microbiota?

2) The key remark previously made is: "Bowel movement frequency is easily assessed". As such it is of importance for primary care. The relevance of the associations between bowel movement frequency and disease lies in: a) The patient can understand that his or her inhabitual bowel movement frequency may form a warning sign for not yet detected disease or for risk of future disease. b) Health authorities could formulate a health promotion message such as for example: "Inform your doctor if, despite a healthy diet, bowel movement frequency remains higher than once daily or lower than once every three days". This forms the relevance for patients of the two papers on bowel movement frequency and disease of the authors. c) When a doctor is confronted with a patient with inhabitual bowel movement frequency she or he should interpret this as a possible warning sign for not yet detected colorectal cancer. The doctor could use this information in the context of an existing algorithm for identifying patients with possible colorectal cancer on the basis of additional parameters including: family history of gastrointestinal cancer, alcohol consumption, anaemia, rectal bleeding, abdominal pain,

appetite loss, and weight loss (2). The algorithm will detect people at a higher risk for which further examination would be appropriate. Patients without these additional characteristics could be warned for the risk of future diseases as identified by the authors in the present paper. Such warnings need to be accompanied by recommendations for prevention (see later comments on dietary factors etc.). This reasoning should be described in part in the rationale for the study in the introduction section and another part in the discussion, the conclusion and the abstract. The remark of the authors: "Considering different underlying mechanisms and duplicate publication, we did not show the result of CRC in the present paper", is not adequate in the present context for two reasons: a) relevance for patients and clinicians can only be clarified if the results are discussed in coherence. b) Results published in the Chinese language are not accessible to a majority of the readers of BMJ Open.

3) In their discussion the authors mention a number of factors that have been shown by others to be relevant for prevention of non-communicable diseases including: microbiota, oxidative stress and dietary habits. All three of these influence bowel movement frequency too. They also influence each other. As an example dietary habits influence the microbiota (3) and oxidative stress (4), microbiota influence oxidative stress as mentioned by the authors. Among dietary habits, consumption of fruit is important for prevention of a number of diseases as shown by the authors in previous studies. Limiting the consumption of saturated fat is also important for prevention of disease. Fruit consumption is important for a proper functioning of the gastrointestinal tract (5). Others have shown that consumption of saturated fat is associated with constipation, even stronger so in diabetic patients (6). All these factors and their interactions play roles in the mechanisms that explain the associations between bowel movement frequency and health issues. Crucial is that bowel movement frequency is an indicator of intestinal health and that this is predictive for the health status of people. In case bowel transit is disturbed it can in a number of people be improved by dietary measures of which it has been proven that they help prevent disease.

4) The authors were so kind to show in their response letter that the proportion of people with daily bowel movements increases when more fruit is consumed! (Graph 1). Fruit consumption appears to normalise bowel movement frequency as is not unexpected: "Gelling fibers soften hard stool in constipation, firm loose/liquid stool in diarrhea, and normalize stool form in patients with IBS" (7). The authors collected data on vegetable consumption too. What about the association of combined fruit and vegetable consumption with bowel movement frequency? What about the effect of consumption of saturated fat? The authors have the information on meat consumption! Diet plays an important role in modifying bowel movement frequency and in modifying health outcomes at least in a significant number of patients. This should be made clear in the paper if one wants to describe the value of the research for patients and

clinicians. Graphs like graph 1 in the response letter should be improved and included in the article. The results in graph 2 in the response letter do not show a difference in bowel movement frequency between the north and south of China. I should perhaps have been more precise in my previous advice. I do not know the geography of China very well yet. Hyperhomocysteinemia has been claimed to be related to insufficient consumption of fruit and vegetables (and thus of folate) and to higher incidence of vascular disease in some rural provinces in northern China. For this reason I asked the authors about bowel movement frequency in the provinces with hyperhomocysteinemia as compared to those with normal levels. It is a second approach to verify the importance of diet for health outcomes and for bowel movement frequency as an indicator at a population level. Such data would be of direct relevance for public health in China which is the goal of the Kadoorie studies (1). A relevant public health message could directly address the citizen: "Eat more fruit and vegetables in case your bowel movement frequency is too high or too low". The message would allow, a personalised approach on the basis of a parameter that the citizen can measure without medical or other assistance. I invite the authors to look at Figure 2: "Provincial distribution pattern of pooled prevalence of hyperhomocysteinemia in China", in the article: Prevalence of hyperhomocysteinemia in China: a systematic review and meta-analysis (8). I copied it below, second image. This demonstrates that the distribution of north/south provinces that the authors choose (see figure 1) North: Qingdao, Harbin, Ganzu and Henan. South: Suzhou, Zhejiang, Sichuan, Hunan, Liuzhou, and Haikou may not represent provinces with high respectively normal homocysteine levels. A more relevant result could be obtained by comparing provinces with established high and normal homocysteine levels (8). Could you please compare for example Henan with Sichuan, or Henan with Qingdao? If the study would be underpowered for that comparison one could compare results for Henan, Suzhou and Zhejiang with those for Sichuan and Qingdao.

5) In their work the authors sometimes use the word constipation for low frequency bowel movements. While this is tempting and while it has been done by others it is not completely correct. The paper would be stronger if internationally used definitions of constipation (Rome criteria) would be at least mentioned.

6) Although the paper is very well written in good English some minor editing may still be desirable.

7) It remains an enigma why the authors have not published any data on the incidence of other cancers besides the colorectal ones in relation to bowel movement frequency.

8) As soon as the authors show how their present data can contribute to specific and cost-effective measures for public health they might find it easier to receive the funds they need for expanding their work to the study of the blood samples.

	Conclusions: The authors should rewrite the introduction, the discussion, the conclusions and the abstract. The rationale for the study in the introduction should be better described and should be coherent with the goals of the Kadoorie Biobank studies earlier described. The effect of dietary factors on bowel movement frequency should be shown and the effect on prevention of disease discussed. The value of the results (including the results on colorectal cancer) for patients, for health authorities and for clinicians should be described as outlined above. The results of the influence of dietary factors on bowel movement frequency need to be further evaluated as outlined above and taken into account in the paper. Without these results patients and clinicians may learn how to detect risk for disease but they are not offered a rationale for prevention. References 1) Epidemiology and the control of disease in China, with emphasis on the Chinese Biobank Study. Li L, Guo Y, Chen Z, Chen J, Peto R. Public Health. 2012 Mar;126(3):210-213. PMID: 22325671 2) Identifying patients with suspected colorectal cancer in primary care: derivation and validation of an algorithm. Hippisley-Cox J, Coupland C. Br J Gen Pract. 2012 Jan;62(594):e29-37. PMID: 22520670 3) Long-term Paleolithic diet is associated with lower resistant starch intake, different gut microbiota composition and increased serum TMAO concentrations. Genoni A, Christophersen CT, Lo J, Coghlan M, Boyce MC, Bird AR, Lyons-Wall P, Devine A. Eur J Nutr. 2019 Jul 5. PMID: 31273523 4) Evolutionary-Concordance Lifestyle and Diet and Mediterranean Diet Pattern Scores and Risk of Incident Colorectal Cancer in Iowa Women. Cheng E, Um CY, Prizment AE, Lazovich D, Bostick RM. Cancer Epidemiol Biomarkers Prev. 2018 Oct;27(10):1195-1202. PMID: 30108096 5) Whole Fruits and Fruit Fiber Emerging Health Effects. Dreher ML. Nutrients. 2018 Nov 28;10(12). pii: E1833. doi: 10.3390/nu10121833. Review. PMID: 30487459
--	--

	6) Association of high dietary saturated fat intake and uncontrolled diabetes with constipation: evidence from the National Health and Nutrition Examination Survey. Taba Taba Vakili S, Nezami BG, Shetty A, Chetty VK, Srinivasan S. Neurogastroenterol Motil. 2015 Oct;27(10):1389-97. PMID: 26176421 7) Fiber supplements and clinically proven health benefits: How to recognize and recommend an effective fiber therapy. Lambeau KV, McRorie JW Jr. J Am Assoc Nurse Pract. 2017 Apr;29(4):216-223. Review. PMID: 28252255 8) Prevalence of hyperhomocysteinemia in China: a systematic review and meta-analysis. Yang B, Fan S, Zhi X, Wang Y, Wang Y, Zheng Q, Sun G. Nutrients. 2014 Dec 29;7(1):74-90. Review. PMID: 25551247
--	--

VERSION 2 – AUTHOR RESPONSE

Reviewer 2: Dr. Keiichi Sumida

Comment: The manuscript has been much improved after revision. Just one comment: Although the authors responded that kidney disease is not regarded as a confounder, given the association of CKD with bowel movement frequency (Toxins. May 16 2018;10(5)) and with outcomes, it should be recognized as a confounder.

Response: We thank Dr. Sumida for your encouraging comment and kind reminder. We have tried additionally adjusting for kidney disease at baseline in the Cox regression models (*page 9: line 18-19*), except for the analysis of chronic kidney diseases. The results almost remained unchanged (data not shown). Considering the previous comment of Dr. Sumida, we have added the prevalence of COPD and CKD into Table 1 (*page 23*).

Reviewer 3: Prof. Alphons J.M. Vermorken

General comments:

The authors have done a rapid and efficient job in their efforts to take each remark of the reviewers into account. I am grateful for their work.

Nevertheless it is not always possible to deal with an issue in the manuscript by adding a sentence in the original text only. A few issues therefore remain to be discussed.

The authors state that: "A properly functioning gastrointestinal tract plays an essential role in health". They argue that bowel movement frequency may represent a simple quantifiable indicator of adequate colonic function. This is correct and for this reason associations between bowel movement frequency and incidence of disease might be used for

the purpose of prevention and early detection of disease. Providing scientific evidence for prevention and control of non-communicable diseases has been declared to be the goal of the Kadoorie Studies as published previously by a team including some of the present authors. Nevertheless the authors fail to describe how their data can contribute to their own goal.

Conclusions:

The authors should rewrite the introduction, the discussion, the conclusions and the abstract.

The rationale for the study in the introduction should be better described and should be coherent with the goals of the Kadoorie Biobank studies earlier described.

The effect of dietary factors on bowel movement frequency should be shown and the effect on prevention of disease discussed.

The value of the results (including the results on colorectal cancer) for patients, for health authorities and for clinicians should be described as outlined above.

The results of the influence of dietary factors on bowel movement frequency need to be further evaluated as outlined above and taken into account in the paper. Without these results patients and clinicians may learn how to detect risk for disease but they are not offered a rationale for prevention.

Response: We thank Prof. Vermorken again for providing us more detailed comments to make this manuscript better and more meaningful. We have responded to these comments one by one as below.

(NOTE: The reference in the original comments is omitted.)

Comment 1: BMJ Open states that it focusses on research that is relevant to patients and clinicians. The present paper (still) fails to spell out its relevance for patients and clinicians. The authors state in their response letter that: "the purpose of the present study was to investigate the associations of bowel movement frequency with multiple health outcomes". That is the purpose for the epidemiologist. The relevance of these associations for patients and clinicians have, however, to be discussed in the relevant context in which previous papers of the same authors also play a role. While in the first manuscript a not completely correct effort was made to describe the relevance in the "Conclusions" this paragraph has now been modified without providing a "message" for patients or clinicians.

Moreover, it is unclear how the data obtained can lead to the second part of the new statement: "Our findings highlight that gastrointestinal health is related to the health of multiple systems and provide preliminary epidemiological evidence for potential systemic effects of gut microbiota". The microbiota were discussed as a relevant factor and others have shown that it plays a role but do the results of the present paper provide evidence for the involvement of the microbiota?

Response: We have recognized that the revision for the conclusion in the last manuscript may miss messages for patients and clinicians. We also agree that our findings might not be able to support the hypothesis about gut microbiota directly. We have revised our conclusion accordingly (*page 17: line 17-21; page 18: 1-5*).

Comment 2: The key remark previously made is: "Bowel movement frequency is easily assessed". As such it is of importance for primary care. The relevance of the associations between bowel movement frequency and disease lies in: a) The patient can understand that his or her inhabitual bowel movement frequency may form a warning sign for not yet detected disease or for risk of future disease. b) Health authorities could formulate a health promotion message such as for example: "Inform your doctor if, despite a healthy diet, bowel movement frequency remains higher than once daily or lower than once every three days". This forms the relevance for patients of the two papers on bowel movement frequency and disease of the authors. c) When a doctor is confronted with a patient with inhabitual bowel movement frequency she or he should interpret this as a possible warning sign for not yet detected colorectal cancer. The doctor could use this information in the context of an existing algorithm for identifying patients with possible colorectal cancer on the basis of additional parameters including: family history of gastrointestinal cancer, alcohol consumption, anaemia, rectal bleeding, abdominal pain, appetite loss, and weight loss (2). The algorithm will detect people at a higher risk for which further examination would be appropriate. Patients without these additional characteristics could be warned for the risk of future diseases as identified by the authors in the present paper. Such warnings need to be accompanied by recommendations for prevention (see later comments on dietary factors etc.). This reasoning should be described in part in the rationale for the study in the introduction section and another part in the discussion, the conclusion and the abstract.

The remark of the authors: "Considering different underlying mechanisms and duplicate publication, we did not show the result of CRC in the present paper", is not adequate in the present context for two reasons: a) relevance for patients and clinicians can only be clarified if the results are discussed in coherence. b) Results published in the Chinese language are not accessible to a majority of the readers of BMJ Open.

Response: We think Prof. Vermorken has pointed out the potential directions for the clinical practice in the future based on our findings. Deeply inspired by the views of Prof. Vermorken above, we have revised our abstract (*page 2: line 2-14, page 3: line 4-6*), introduction (*page 4: line 2-21; page 5: line 1-2*), discussion and conclusion (*page 17: line 17-21; page 18: 1-5*), respectively.

As for the results of colorectal cancer (CRC) that we have published in *Chin J Epidemiol*¹, we are sorry for not including them in this manuscript. The academic community of China especially emphasizes to avoid this kind of duplicate publication in different languages in more recent years. We would be regarded as academic misconduct if we do so. Alternatively, we have cited this paper in the introduction section (*page 4: line 13-16*) and the conclusion section (*page 17: line 21; page 18: line 1*). Also, the abstract in English of this paper is available at PubMed (PMID: 31006195). We would happy to communicate with anyone who may be interested in that CRC paper in Chinese.

Comment 3: In their discussion the authors mention a number of factors that have been shown by others to be relevant for prevention of non-communicable diseases including: microbiota, oxidative stress and dietary habits. All three of these influence bowel movement frequency too. They also influence each other. As an example dietary habits influence the microbiota (3) and oxidative stress (4), microbiota influence oxidative stress as mentioned by the authors. Among dietary habits, consumption of fruit is important for prevention of a number of diseases as shown by the authors in previous studies. Limiting the consumption of saturated fat is also important for prevention of disease. Fruit consumption is important for a proper functioning of the gastrointestinal tract (5). Others have shown that consumption of saturated fat is associated with constipation, even stronger so in diabetic patients (6). All these factors and their interactions play roles in the mechanisms that explain the associations between bowel movement frequency and health issues. Crucial is that bowel movement frequency is an indicator of intestinal health and that this is predictive for the health status of people. In case bowel transit is disturbed it can in a number of people be improved by dietary measures of which it has been proven that they help prevent disease.

Response: We agree that many factors might affect bowel movement frequency (BMF). In the present analyses, we concentrated on the associations of BMF with multiple diseases. As long as such an association between BMF and health is established, the associations of lifestyle factors, including but not limited to dietary habits, with BMF and strategies for maintaining normal BMF are warranted for further research. Unfortunately, the current manuscript has no space to include more analysis due to word limits. We would happy to analyse the association between lifestyle factors and BMF soon according to Prof.

Vermorcken's suggestion. Please see our response to **comment 4** for more detailed information.

Comment 4: The authors were so kind to show in their response letter that the proportion of people with daily bowel movements increases when more fruit is consumed! (Graph 1). Fruit consumption appears to normalise bowel movement frequency as is not unexpected: "Gelling fibers soften hard stool in constipation, firm loose/liquid stool in diarrhea, and normalize stool form in patients with IBS" (7). The authors collected data on vegetable consumption too. What about the association of combined fruit and vegetable consumption with bowel movement frequency? What about the effect of consumption of saturated fat? The authors have the information on meat consumption! Diet plays an important role in modifying bowel movement frequency and in modifying health outcomes at least in a significant number of patients. This should be made clear in the paper if one wants to describe the value of the research for patients and clinicians. Graphs like graph 1 in the response letter should be improved and included in the article.

The results in graph 2 in the response letter do not show a difference in bowel movement frequency between the north and south of China. I should perhaps have been more precise in my previous advice. I do not know the geography of China very well yet. Hyperhomocysteinemia has been claimed to be related to insufficient consumption of fruit and vegetables (and thus of folate) and to higher incidence of vascular disease in some rural provinces in northern China. For this reason I asked the authors about bowel movement frequency in the provinces with hyperhomocysteinemia as compared to those with normal levels. It is a second approach to verify the importance of diet for health outcomes and for bowel movement frequency as an indicator at a population level. Such data would be of direct relevance for public health in China which is the goal of the Kadoorie studies (1). A relevant public health message could directly address the citizen: "Eat more fruit and vegetables in case your bowel movement frequency is too high or too low". The message would allow, a personalised approach on the basis of a parameter that the citizen can measure without medical or other assistance. I invite the authors to look at Figure 2: "Provincial distribution pattern of pooled prevalence of hyperhomocysteinemia in China", in the article: Prevalence of hyperhomocysteinemia in China: a systematic review and meta-analysis (8). I copied it below, second image. This demonstrates that the distribution of north/south provinces that the authors choose (see figure 1) North: Qingdao, Harbin, Ganzu and Henan. South: Suzhou, Zhejiang, Sichuan, Hunan, Liuzhou, and Haikou may not represent provinces with high respectively normal homocysteine levels. A more relevant result could be obtained by comparing provinces with established high and normal homocysteine levels (8). Could you please compare for example Henan with Sichuan, or Henan with Qingdao? If the study would

be underpowered for that comparison one could compare results for Henan, Suzhou and Zhejiang with those for Sichuan and Qingdao.

Response: As we have discussed in the response to **comment 3**, this manuscript did not aim at investigating all the factors that might affect bowel movement frequency (BMF). Also, we have acknowledged that the dietary frequency questionnaire used in our study was limited. We only asked participants about the five categories of frequency (daily, 4 to 6 days per week, 1 to 3 days per week, monthly, or never or rarely) of 12 major food groups during the previous 12 months at baseline survey; the amounts of consumption per day were not collected.

The following **Table A** shows the distributions of BMF in different dietary frequency groups. Due to a lack of information on consumption amounts, there were only minor differences in the BMF between different consumption frequency groups. We think that more valuable information might be obtained by investigating the association between diet and BMF in studies with a more detailed dietary assessment. We would prefer not to add these results to the current manuscript.

According to the systematic review that Prof. Vermorken has referred², we have compared the distribution of BMF in ten diverse regions separately (**Table B**). We found the results were not in favor of the hypothesis based on the prevalence of hyperhomocysteinemia. For example, the pooled prevalence of hyperhomocysteinemia in Henan was the highest according to Yang B, et al², while the proportion of participants who had bowel movements daily in Henan (73.1%) was similar to those in Zhejiang (73.6%) and Sichuan (71.8%). These results indicate the complexity of the affecting factors of BMF. Considering the relevance to the topic of this paper and potential ambiguity, we would also prefer not to show them in the manuscript.

Table A. Distribution of BMF in different dietary frequency groups

	No. of participants	More than once on most days (%)	About daily (%)	Once every 2-3 days (%)	Less than 3 times a week (%)
Fresh fruit					
Daily	88 929	9.1	78.8	8.7	3.3
4-6 d/wk	45 985	9.2	76.9	9.8	4.1

1-3 d/wk	154 509	9.5	76.5	9.7	4.3
Monthly	167 073	9.5	76.2	9.4	4.8
Never/rarely	30 702	10.9	70.7	11.5	6.9
Fresh vegetable					
Daily	461 312	9.6	76.5	9.6	4.3
4-6 d/wk	17 480	8.6	77.5	9.6	4.3
1-3 d/wk	6924	8.3	78.3	8.7	4.7
Monthly	1362	6.6	85.4	3.8	4.2
Never/rarely	120	6.9	68.8	18.3	6.0
Fresh fruit and vegetable combined					
Both daily	87 957	9.2	78.8	8.7	3.3
Either daily	374 327	9.6	75.9	9.8	4.6
Neither daily	24 914	8.3	77.8	9.3	4.6
Meat					
Daily	141 464	9.8	76.7	9.4	4.1
4-6 d/wk	88 639	8.9	77.6	9.4	4.0
1-3 d/wk	173 523	9.4	76.8	9.5	4.4
Monthly	60 841	9.8	75.8	9.6	4.8
Never/rarely	22 731	11.1	71.6	11.2	6.0

BMF: bowel movement frequency. Age (continuous), gender (male, female) and region (10 sites) were adjusted in the multinomial logistic regression model.

Table B. Distribution of BMF in ten diverse regions

Region	No. of participants	More than once on most days (%)	About daily (%)	Once every 2-3 days (%)	Less than 3 times a week (%)	prevalence of HHcy (%)*
Gansu (R, N, I)	48 469	6.2	84.1	7.4	2.2	No data
Liuzhou (U, S, C)	46 407	9.5	80.5	7.4	2.7	No data
Haikou (U, S, C)	28 940	3.8	79.9	12.6	3.7	0-25
Qingdao (U, N, C)	33 354	6.4	78.7	11.2	3.7	0-25
Harbin (U, N, I)	49 516	7.6	77.3	9.1	6.1	No data
Hunan (R, S, I)	56 998	15.0	76.5	6.0	2.4	No data
Suzhou (U, S, C)	51 919	8.0	74.9	9.6	7.5	25-35
Zhejiang (R, S, C)	56 728	12.4	73.6	10.1	3.9	25-35
Henan (R, N, I)	59 866	13.4	73.1	9.8	3.7	>45
Sichuan (R, S, I)	55 001	7.5	71.8	13.9	6.8	0-25

BMF: bowel movement frequency. R, rural; U, urban; N, north; S, south; I, inland; C, coastal.
HHcy, hyperhomocysteinemia

Age (continuous) and gender (male, female) were adjusted in the multinomial logistic regression model.

*Data came from Yang B, et al.²

Comment 5: In their work the authors sometimes use the word constipation for low frequency bowel movements. While this is tempting and while it has been done by others it is not completely correct. The paper would be stronger if internationally used definitions of constipation (Rome criteria) would be at least mentioned.

Response: We have replaced the word “constipation” used in our study by “low bowel movement frequency” (*page 13: line 5; page 14: line 21*). The word “constipation” used by previously published studies was kept.

Comment 6: Although the paper is very well written in good English some minor editing may still be desirable.

Response: We thank Prof. Vermorken for your encouragement and kind reminder. This manuscript has been carefully edited by a colleague with an excellent mastery of the English language. The changes were highlighted by using the track changes mode in the “Main Document – marked copy”.

Comment 7: It remains an enigma why the authors have not published any data on the incidence of other cancers besides the colorectal ones in relation to bowel movement frequency.

Response: We determine the research objectives mainly based on available evidence from the previous laboratory or population studies and potential mechanisms underlying the associations, as most researchers do. Research into the associations of BMF with different health outcomes is still underway. In the future, with the extension of follow-up time and the increase in the number of accumulated cases, we would be able to investigate the association between BMF and the health outcomes with low incidence.

Comment 8: As soon as the authors show how their present data can contribute to specific and cost-effective measures for public health they might find it easier to receive the funds they need for expanding their work to the study of the blood samples.

Response: We thank Prof. Vermorken for this critical suggestion. We are making efforts to achieve this goal.

References

1. Yang S, Shen Z, Yu C, et al. Association between the frequency of bowel movements and the risk of colorectal cancer in Chinese adults. *Chin J Epidemiol* 2019;40(4):382-88. doi:10.3760/cma.j.issn.0254-6450.2019.04.003
2. Yang B, Fan S, Zhi X, et al. Prevalence of hyperhomocysteinemia in China: a systematic review and meta-analysis. *Nutrients* 2014;7(1):74-90. doi:10.3390/nu7010074

VERSION 3 – REVIEW

REVIEWER	A.J.M. Vermorken Northwest University, College of Life Sciences
REVIEW RETURNED	08-Dec-2019

GENERAL COMMENTS	The paper has been improved and I recommend that it is accepted.
--